EMBO
*reports*

# *Streptococcus pneumoniae* hijacks host autophagy by deploying CbpC as a decoy for Atg14 depletion

Sayaka Shizukuishi[1,2], Michinaga Ogawa[1,*] (iD), Satoko Matsunaga[2], Mikado Tomokiyo[1,3], Tadayoshi Ikebe[1], Shinya Fushinobu[4,5] (iD), Akihide Ryo[2] & Makoto Ohnishi[1]

## Abstract

Pneumococcal cell surface-exposed choline-binding proteins (CBPs) play pivotal roles in multiple infectious processes with pneumococci. Intracellular pneumococci can be recognized at multiple steps during bactericidal autophagy. However, whether CBPs are involved in pneumococci-induced autophagic processes remains unknown. In this study, we demonstrate that CbpC from *S. pneumoniae* strain TIGR4 activates autophagy through an interaction with Atg14. However, *S. pneumoniae* also interferes with autophagy by deploying CbpC as a decoy to cause autophagic degradation of Atg14 through an interaction with p62/SQSTM1. Thus, *S. pneumoniae* suppresses the autophagic degradation of intracellular pneumococci and survives within cells. Domain analysis reveals that the coiled-coil domain of Atg14 and residue Y83 of the dp3 domain in the N-terminal region of CbpC are crucial for both the CbpC–Atg14 interaction and the subsequent autophagic degradation of Atg14. Although homology modeling indicates that CbpC orthologs have similar structures in the dp3 domain, autophagy induction through Atg14 binding is an intrinsic property of CbpC. Our data provide novel insights into the evolutionary hijacking of host-defense systems by intracellular pneumococci.

**Keywords** Atg14; autophagy; CbpC; p62; *Streptococcus pneumoniae*
**Subject Categories** Autophagy & Cell Death; Microbiology, Virology & Host Pathogen Interaction; Signal Transduction

## Introduction

*Streptococcus pneumoniae* is a major, encapsulated gram-positive pathogen that causes diseases including community-acquired pneumonia, meningitis, and sepsis [1,2]. During severe infections, *S. pneumoniae* colonization of nasopharyngeal epithelial cells can lead to epithelial barrier penetration and entrance into the bloodstream and brain via the blood–brain barrier [1,2]. Although multivalent pneumococcal polysaccharides and conjugate vaccines are available and generally effective, they also have major shortcomings with respect to the emergence of vaccine-resistant serotypes (serotype replacement) [2,3]. The increasing prevalence of antibiotic-resistant pneumococci is a global problem [3]. Therefore, alternative therapeutic approaches are needed. However, progress has been limited by an incomplete understanding of virulence factors and the intracellular fate of *S. pneumoniae*.

Pneumococcal cell surface proteins, including LPXTG motif-containing proteins, lipoproteins, and choline-binding proteins (referred to as CBPs or Cbps), are key pathogenic factors [4–8] and are viewed as primary therapeutic targets [7]. *S. pneumoniae* has an absolute nutritional requirement for choline. Its characteristic cell wall is composed of lipoteichoic and teichoic acids and is decorated with phosphocholine (PCho) [4,7]. *S. pneumoniae* have more than 15 CBPs, with PCho acting as a scaffold for all of them at the cell wall (Fig 1A) [4,7]. All CBP family proteins share choline-binding modules (CBMs) composed of choline-binding repeats (CBRs), which facilitate their binding to the cell wall [4,7]. To date, the crystal structures of 7 pneumococcal CBM-containing proteins have been solved, including CbpE [9], CbpF [10], CbpJ [11], CbpL [12], and LytA [13]. Although it has been reported that CBPs are involved in pathogenic functions of pneumococci, including adhesion to host cells, bacterial autolysis, and complement activation, the functional understanding of CBPs is incomplete [4–8].

Xenophagy functions as an innate host-defense system against microbial intruders, providing a first line of defense [14–16]. Upon intracellular pathogen invasion, xenophagy is activated in host cells by the recognition of bacterial components or infection processes via multiple cytosolic sensors [14]. However, many intracellular bacterial pathogens have evolved strategies to subvert xenophagy, including evading autophagic recognition, dampening autophagosome formation, and manipulating autophagosome–lysosome fusion [14,17–19]. Previously, we showed that intracellular *S. pneumoniae* can be recognized by bactericidal autophagy [20]; however, whether and how these autophagic processes are manipulated by pneumococcal virulence factors are mostly unknown.

1 Department of Bacteriology I, National Institute of Infectious Diseases, Tokyo, Japan
2 Department of Microbiology, Yokohama City University Graduate School of Medicine, Kanagawa, Japan
3 School of Veterinary Medicine, Azabu University, Kanagawa, Japan
4 Department of Biotechnology, Graduate School of Agricultural and Life Sciences, The University of Tokyo, Tokyo, Japan
5 Collaborative Research Institute for Innovative Microbiology, The University of Tokyo, Tokyo, Japan
*Corresponding author. Tel: +81 3 5285 1111; E-mail: micogawa@nih.go.jp

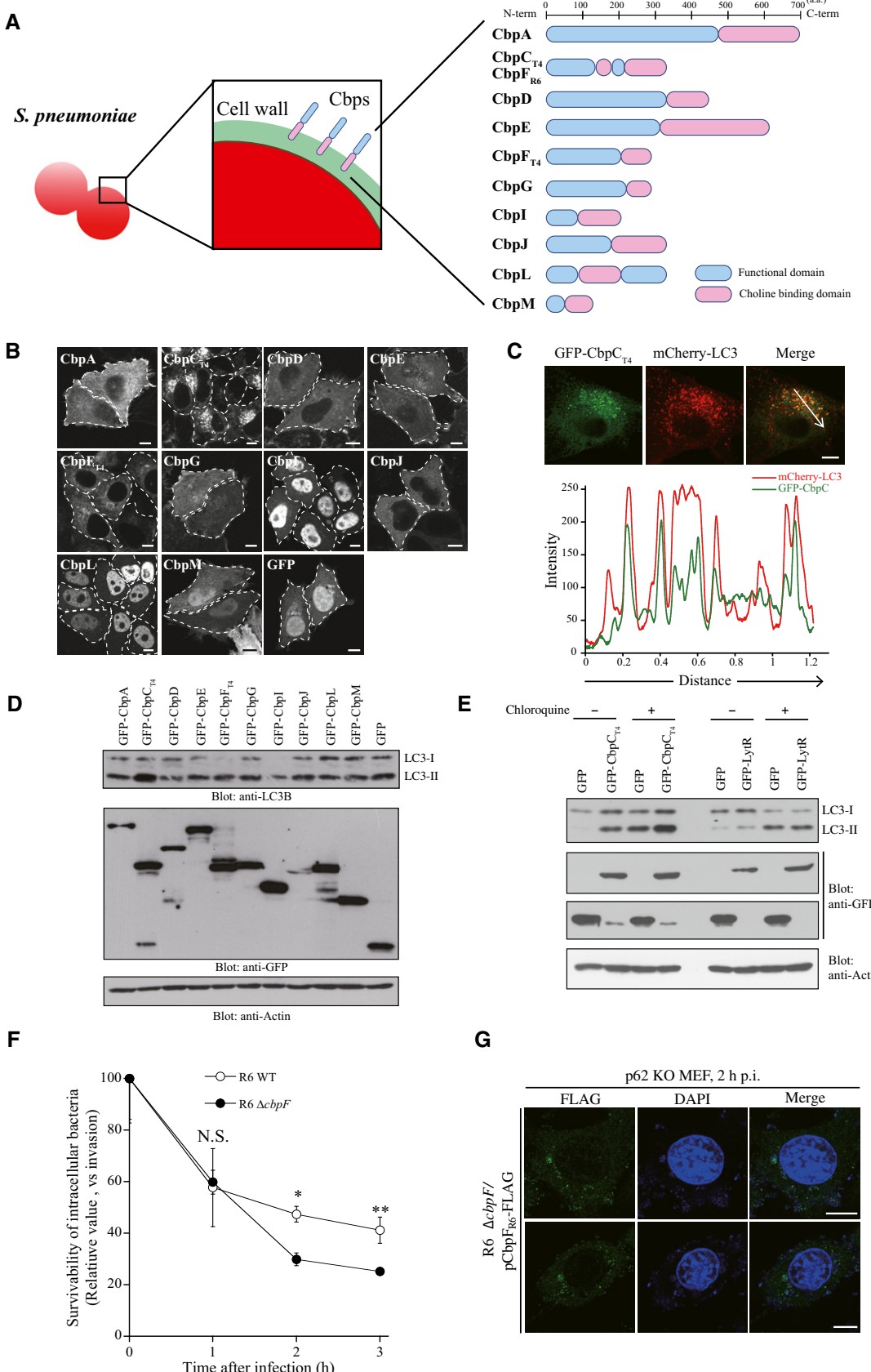

Figure 1.

◀

**Figure 1.  The pneumococcal CbpC protein can activate autophagy, but can also facilitate intracellular pneumococcal survival.**

A  Pneumococcal CBPs used in (B).
B  Confocal images of HeLa cells transiently expressing GFP-CBPs or GFP. The dotted lines show each cell shape. Scale bars, 10 μm.
C  Confocal images of HeLa cells transiently expressing GFP-CbpC$_{T4}$ and mCherry-LC3 (upper). The fluorescence intensities of GFP-CbpC$_{T4}$ (green) and mCherry-LC3 (red) along the arrow are shown in the graph at the bottom.
D  Lysates from 293T cells transiently expressing GFP-Cbps or GFP were subjected to SDS–PAGE and analyzed by immunoblotting using antibodies against LC3, GFP, or actin.
E  Lysates from 293T cells transiently expressing GFP-CbpC$_{T4}$, GFP-LytR, or GFP in the presence or absence of chloroquine were subjected to SDS–PAGE and analyzed by immunoblotting using antibodies against LC3, GFP, or actin.
F  MEFs were infected with *S. pneumoniae* R6 WT or Δ*cbpF* for the indicated periods, and the intracellular survival of bacteria expressed as the number of CFUs.
G  p62-KO MEF cells infected with *S. pneumoniae* R6 Δ*cbpF*/pCbpF$_{R6}$-FLAG for 2 h were fixed and stained with DAPI and an anti-FLAG antibody, and representative epifluorescence images are shown. Scale bars, 10 μm.

Data information: In (F), data represent mean ± SEM of 3 biological replicates. Student's *t*-test was used to calculate statistical significance. *$P < 0.05$, **$P < 0.01$.
Source data are available online for this figure.

In this study, we demonstrated that CbpC from *S. pneumoniae* strain TIGR4 induces autophagy by interacting with Atg14. Our data also revealed that the p62–CbpC–Atg14 complex causes the selective autophagy targeting Atg14, which eventually attenuates the autophagic degradation of intracellular pneumococci.

## Results

### The pneumococcal CbpC protein can activate autophagy and also facilitate intracellular pneumococcal survival

*Streptococcus pneumoniae* undergoes spontaneous autolysis during infection, during which pneumococcal cell wall-associated components, including CBPs and LPXTG proteins, diffuse into the cytosol of host cells through endosomal membrane pores formed by pneumolysin, a cholesterol-binding cytolysin [21]. Therefore, we investigated the intracellular functions of CBPs in pneumococcal virulence (Fig 1A). In this study, we used the nomenclature from the *S. pneumoniae* TIGR4 strain. When Cbps such as CbpA, C, D, E, F, G, I, J, L, and M were ectopically expressed in HeLa cells as green fluorescence protein (GFP)-fusion proteins, we found that GFP-CbpC (CbpC from TIGR4, hereafter referred as CbpC$_{T4}$) caused the formation of intracellular inclusion bodies, reminiscent of autophagic puncta (Fig 1B). CbpC$_{T4}$ is one of the most abundant proteins in the pneumococcal cell wall [10,22]. Upon transient expression in HeLa cells, GFP-CbpC$_{T4}$ co-localized with mCherry-LC3, an intrinsic autophagosome marker [23] (Fig 1C). The amount of LC3-II (a membrane-bound form of LC3) exclusively increased in cells expressing GFP-CbpC$_{T4}$ (Figs 1D and EV1A). To determine whether the increase in LC3-II and GFP-LC3-positive puncta was caused by autophagy activation or inhibited autophagic degradation, we performed autophagic flux assays in chloroquine-treated, GFP-CbpC$_{T4}$-expressing cells. Upon chloroquine treatment, GFP-CbpC$_{T4}$-induced LC3-II accumulation was further augmented, indicating that transiently expressed CbpC$_{T4}$ was involved in activating autophagy (Fig 1E).

To investigate the physiological role of CbpC in *S. pneumoniae* infection, we conducted intracellular-survivability assays using *S. pneumoniae* R6 or TIGR4. CbpF$_{R6}$ (CbpF from R6, hereafter referred as CbpF$_{R6}$) is a CbpC$_{T4}$ ortholog with 97% sequence similarity (312/340 amino acids) in the N-terminal functional

region (Fig EV2B and C) [10]. Upon infection with the wild-type (WT) and Δ*cbpF*$_{R6}$ or Δ*cbpC*$_{T4}$ strains derived from *S. pneumoniae* R6 or TIGR4, we found that the survival of Δ*cbpF*$_{R6}$ or Δ*cbpC*$_{T4}$ strains was significantly lower than that of WT bacteria at the later stage of infection, whereas CbpF$_{R6}$ and CbpC$_{T4}$ had no adverse effect on bacterial invasion into host cells (Figs 1F and EV1B and C). Next, we examined whether bacterial autolysis releases free CbpC into the cytosol. Upon infection with *S. pneumoniae* R6 Δ*cbpF*/pCbpF$_{R6}$-FLAG, CbpF$_{R6}$ signals were detected proximal to intracellular pneumococci, but also free in the cytosol, whereas these signals were completely abolished in Δ*cbpF*-infected cells (Figs 1G and EV1D and E).

### Identification of Atg14 as a CbpC-interacting protein

We next investigated the mechanism whereby CbpC$_{T4}$ induces autophagy. We hypothesized that CbpC$_{T4}$ could functionally interact with autophagy-related proteins. Therefore, we conducted comprehensive protein–protein interaction analysis. As a first screening, we synthesized a series of autophagy-related proteins using a wheat cell-free system and then performed pulldown assays. Our data demonstrated that CbpC$_{T4}$ could interact with seven autophagy-related proteins, including Atg5, 12, 14, 16L1, Beclin1, WIP1, and WIPI2 (Figs 2A and EV2A). To exclude candidates with non-specific binding, we performed a second screening by performing glutathione S transferase (GST)-based pulldown assays using 293T cell lysates expressing GFP-fusion proteins with Atg5, 12, 14, 16L1, Beclin1, WIPI1, or WIPI2. As GST-CbpC$_{T4}$ expression inhibited *Escherichia coli* growth, we prepared a plasmid encoding GST-CbpF$_{R6}$ for expression in *E. coli* [10]. Based on the results of the second screen, we focused on Atg5 and Atg14 as CbpC$_{T4}$-interacting candidates (Figs 2B and EV2D). Further immunoprecipitation (IP) assays in 293T cells revealed the specific interaction of CbpC$_{T4}$ with Atg14 (Fig 2C and D). Furthermore, GST-pulldown assays using recombinant GST-CbpF$_{R6}$ and 3Myc-Atg14 corroborated the direct interaction of CbpF$_{R6}$ with Atg14 (Fig 2E). Finally, we confirmed the CbpC–Atg14 interaction in *S. pneumoniae*-infected cells. When MEFs (mouse embryonic fibroblast cells) stably expressing hemagglutinin (HA)-tagged Atg14 (MEFs/HA-Atg14) were infected with *S. pneumoniae* R6 Δ*cbpF*/pCbpF$_{R6}$-FLAG, a CbpF$_{R6}$–FLAG–HA–Atg14 interaction was clearly detected during infection (Fig 2F).

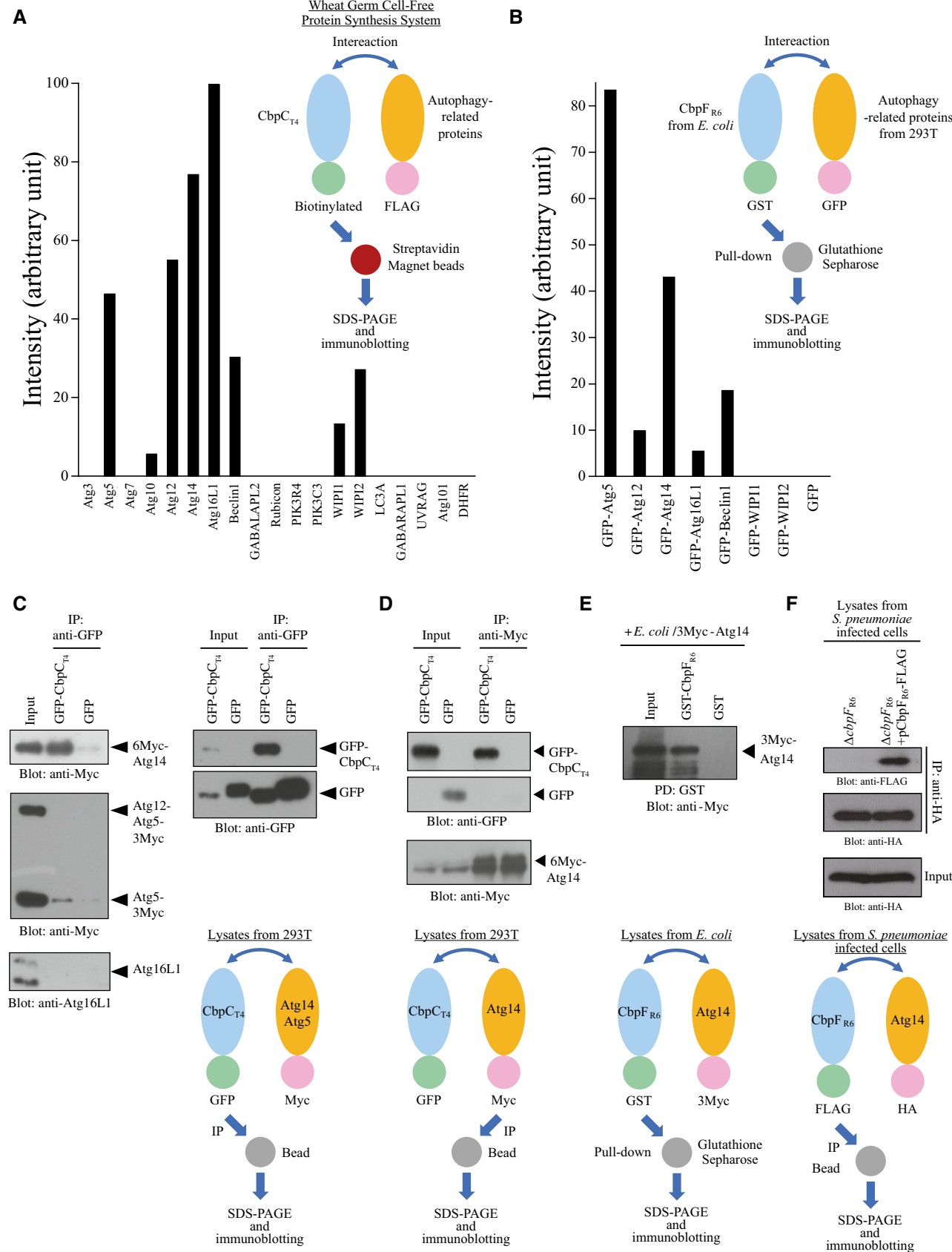

**Figure 2.**

**Figure 2.  Identification of Atg14 as a CbpC-interacting protein.**

A   Quantification of the band intensities following streptavidin-based pulldown assays using the *in vitro*-translated biotinylated CbpC$_{T4}$ and FLAG-tagged autophagy-related proteins shown in Fig EV2A.

B   Quantification of the band intensities following GST-pulldown assays with 293T cell lysates expressing the autophagy-related proteins and recombinant GST-CbpF$_{R6}$ shown in Fig EV2D.

C   Lysates from 293T cells transiently expressing 3Myc-Atg14 or Atg5-3Myc, and GFP-CbpC$_{T4}$ or GFP were immunoprecipitated with GFP-Trap, and the bound proteins were analyzed by immunoblotting.

D   Lysates from 293T cells transiently expressing GFP-CbpC$_{T4}$ or GFP, and 6Myc-Atg14 were immunoprecipitated with an anti-Myc antibody and Protein G PLUS-Agarose. The bound proteins were analyzed by immunoblotting.

E   Recombinant GST-CbpF$_{R6}$ or GST, and 3Myc-Atg14 were used for GST-pulldown assays. The bound proteins were analyzed by immunoblotting using an anti-Myc antibody.

F   MEFs infected with R6 Δ*cbpF* expressing CbpF$_{R6}$-FLAG for 2 h were subjected to IP and assayed using anti-HA agarose beads. The bound proteins were analyzed by immunoblotting using an anti-FLAG antibody. Schematic representations indicating assay designs in (C–F) are shown below the blots.

Source data are available online for this figure.

## CbpC can act as a decoy for autophagic Atg14 degradation and xenophagy subversion

Atg14 plays two roles in autophagy; the first is as a component of the autophagy-specific class III phosphatidylinositol 3-kinase complex through Beclin1-binding [24–26], and the second is as a regulator of autophagosome-lysosome fusion through Stx17 binding [27,28]. Therefore, we hypothesized that *S. pneumoniae* could promote CbpC-driven selective autophagy during early infection and that CbpC–Atg14 binding would lead to Atg14 degradation, which, in turn, would reduce autophagosome–lysosome fusion and bacterial degradation.

To test this hypothesis, we infected MEFs/HA-Atg14 with *S. pneumoniae* TIGR4 WT or Δ*cbpC* in presence or absence of cyclo-heximide (CHX) for 1, 2, or 3 h, and then, we examined whether the amount of HA-Atg14 decreased during early infection in a CbpC-dependent manner. Approximately 1 bacterium was internalized per cell under our experimental conditions, suggesting that most cells were comparably invaded by bacteria (Fig EV3A). Importantly, our findings revealed that the amount of HA-Atg14 dramatically decreased in cells infected with WT bacteria after 2 h of infection, but not in cells infected with Δ*cbpC* bacteria (Fig 3A). We next determined whether autophagic degradation was suppressed in a CbpC-dependent manner by employing p62 as a substrate for autophagic degradation. p62 degradation in CHX-treated cells was dramatically suppressed in cells infected with WT bacteria, while that in Δ*cbpC* bacteria-infected cells and even in uninfected cells rapidly increased (Fig 3A). This result strengthened our hypothesis that CbpC-driven selective autophagy can cause Atg14 degradation, which ultimately suppresses bactericidal autophagy. We then investigated whether a defect in autophagic degradation would prevent the Atg14 degradation induced by *S. pneumoniae* infection. Notably, upon treatment with Bafilomycin A1, a v-ATPase inhibitor, Atg14 degradation in WT bacteria-infected cells was dramatically suppressed, and the Atg14 level was fully restored to that observed in Δ*cbpC* bacteria-infected cells (Fig 3B). To investigate the physiological role of autophagy on the survival of Δ*cbpC*, we conducted intracellular-survivability assays using Atg5-knockout (KO) MEFs, which are autophagy-deficient, and found that Δ*cbpC* survival was comparable to that of WT bacteria (Fig EV3B). These results supported our notion that CbpC-driven autophagy during early infection can cause Atg14 depletion and suppress subsequent bactericidal autophagy. Furthermore, we investigated whether CbpC of

*S. pneumoniae* could affect the abundance or localization of endogenous Atg14 in human lung epithelial cells. Notably, *S. pneumoniae* infection dramatically lowered Golgi-resident Atg14 without affecting the structural integrity of the Golgi in a CbpC-dependent manner (Fig 3C–F). Next, we examined whether Atg14 degradation was due to CbpC release in infected cells. When A549 cells were infected with an invasion-deficient mutant (Δ*cbpA*), an endosomal damage-deficient mutant (Δ*ply*), or a bacterial autolysis-dampened mutant (Δ*lytA*), the disappearance of Atg14 dramatically decreased to a similar level of Δ*cbpF*$_{R6}$ (Fig EV3C and D). These results suggest that bacterial invasion, endosomal damage, and bacterial autolysis play pivotal roles in Atg14 degradation induced by *S. pneumoniae* infection. To further confirm the importance of bacterial invasion for *S. pneumoniae*-induced Atg14 degradation, we constructed a polyclonal *S. pneumoniae* invasion setting using a co-culture system. *S. pneumoniae* invasion-permissive A549 cells (marked with GFP) and *S. pneumoniae* invasion-non-permissive (pIgR knockdown) A549 cells were co-cultured at a 1:3 ratio, and *S. pneumoniae*-induced Atg14 degradation was measured. Robust Atg14 degradation occurred in *S. pneumoniae* invasion-permissive A549 cells (marked with GFP), but not in *S. pneumoniae* invasion-non-permissive (pIgR knockdown) A549 cells, clearly showing that bacterial invasion is essential for Atg14 degradation (Figs 3G and H, and EV3E). Taken together, these results support the idea that *S. pneumoniae* can manipulate autophagy by employing CbpC as a decoy to cause autophagic degradation of Atg14.

We then studied the Atg14-degrading effect of CbpC$_{T4}$ using 293T cells transiently expressing GFP-CbpC$_{T4}$ and HA-Atg14, and we discovered that the amount of HA-Atg14 dramatically decreased by co-expressing GFP-CbpC$_{T4}$ (Fig 3I). To tackle the mechanism of CbpC$_{T4}$-induced Atg14 degradation through selective autophagy, we conducted IP assays using cells co-expressing GFP-CbpC$_{T4}$, p62-3Myc, and HA-Atg14 and found that Atg14–CbpC–p62 complexes could form within cells (Fig 3J). Next, we examined whether Atg5 or p62 knockdown could suppress Atg14 degradation in *S. pneumoniae*-infected A549 cells, and we found that the disappearance of perinuclear Atg14 caused by *S. pneumoniae* infection was dramatically suppressed by p62 knockdown. We also noticed that suppressive effect of Atg5 knockdown on the disappearance of perinuclear Atg14 was not as strong as that observed in p62-knocked down cells, implying the partial involvement of an Atg5-independent degradative pathway, such as the ubiquitin–proteasome system (Figs 3K and EV3E). These results suggested that selective

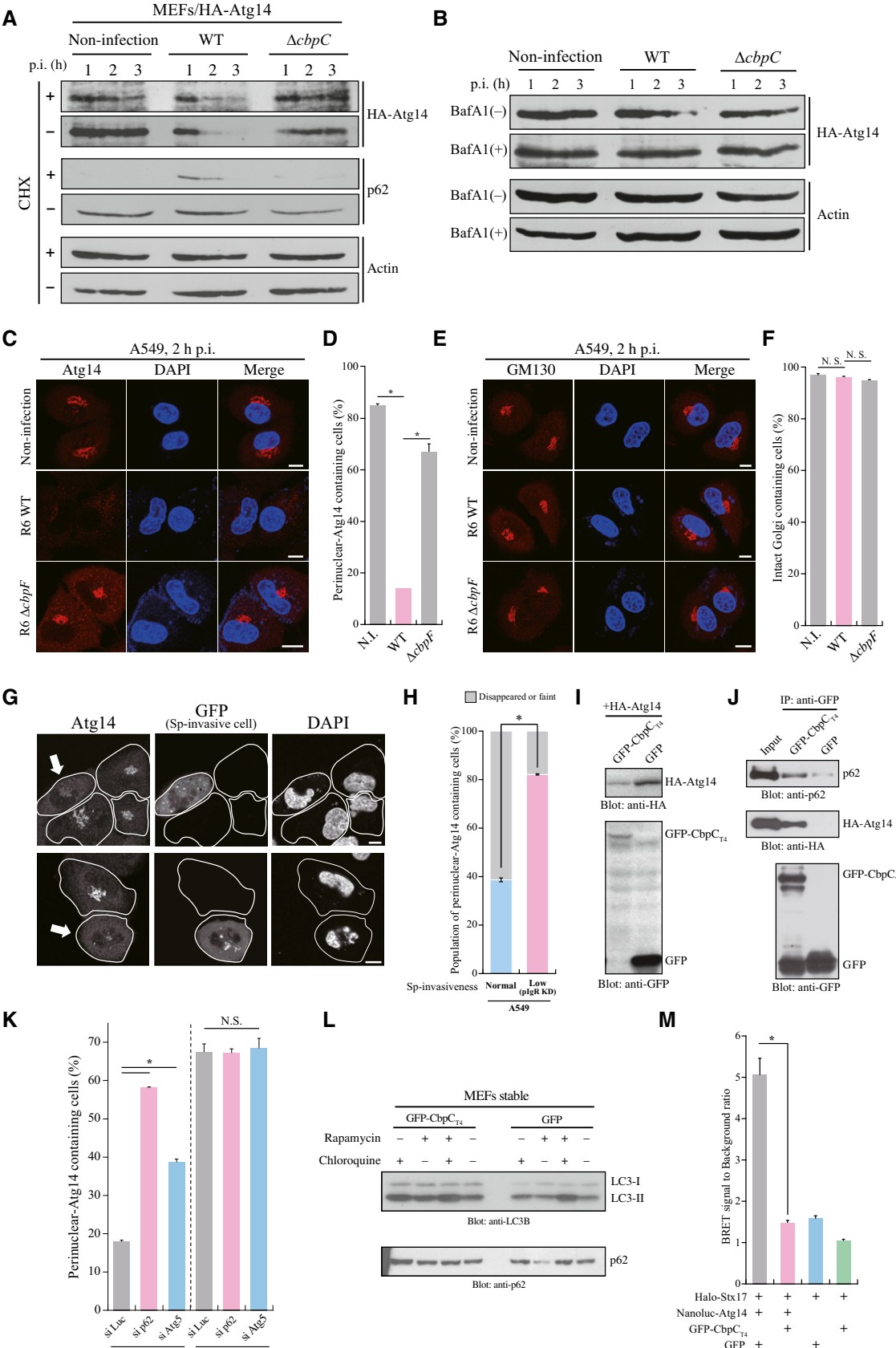

**Figure 3.**

◄

**Figure 3. CbpC could bind to p62 and act as a decoy for autophagic degradation of Atg14 to suppress autophagic degradation.**

A Lysates from MEFs stably expressing HA-Atg14 infected with *S. pneumoniae* TIGR4 WT or Δ*cbpC* for 1, 2, or 3 h in the presence or absence of cycloheximide were subjected to SDS–PAGE and analyzed by immunoblotting with the indicated antibodies.

B MEFs/HA-Atg14 cells were infected with *S. pneumoniae* TIGR4 WT or Δ*cbpC* for 1, 2, or 3 h in the presence or absence of Bafilomycin A1 (BafA1). The lysates were subjected to SDS–PAGE and analyzed by immunoblotting with the indicated antibodies.

C–F (C, E) A549 cells were infected with *S. pneumoniae* R6 WT or Δ*cbpC* for 2 h and fixed and stained with DAPI and an anti-Atg14 or anti-GM130 antibody. Representative epifluorescence images are shown. Scale bars, 10 μm. (D, F) The percentages of perinuclear-localizing Atg14 containing cells in (C) or intact Golgi apparatus-containing cells in (E) were quantified.

G *Streptococcus pneumoniae* (Sp) invasion-permissive A549 cells (marked with GFP) and Sp invasion-non-permissive (pIgR knocked down) A549 cells were co-cultured at a 1:3 ratio. Atg14-disappearance experiments were conducted, and representative epifluorescence images are shown. Scale bars, 10 μm. The lines show each cell shape, and the arrows show invasion-permissive A549 cells marked with GFP.

H The percentages of perinuclear-localizing Atg14 containing cells in (G) were quantified.

I Lysates from 293T cells transiently expressing GFP-CbpC or GFP, and HA-Atg14 were subjected to SDS–PAGE and analyzed by immunoblotting using antibodies against HA or GFP.

J Lysates from 293T cells transiently expressing GFP-CbpC$_{T4}$ or GFP were immunoprecipitated using GST-GFP-Nanobody. Additionally, beads were mixed with lysates from 293T cells transiently expressing p62-3Myc and HA-Atg14, and bound proteins were analyzed by immunoblotting.

K A549 cells treated with the indicated siRNAs were infected with *S. pneumoniae* R6 WT or Δ*cbpF* for the indicated durations. The cells were fixed and stained with DAPI and an anti-Atg14 antibody, and percentages of perinuclear-localizing Atg14 containing cells were quantified.

L Lysates from MEFs stably expressing GFP-CbpC$_{T4}$ or GFP in the presence or absence of rapamycin or chloroquine were subjected to SDS–PAGE and immunoblotted using antibodies against LC3 or p62.

M Quantification of NanoBRET signals in 293A cells transiently expressing HaloTag-Stx17 and Nanoluc-Atg14 in the presence or absence of GFP-CbpC or GFP.

Data information: In (D, F, H, K, M), data represent mean ± SEM of 3 biological replicates. Student's *t*-test was used to calculate statistical significance. \*P < 0.01, N.S., not significant.

Source data are available online for this figure.

autophagy through the Atg14–CbpC–p62 axis during early infection is involved in subsequent Atg14 degradation.

At a later stage of the autophagic process, Atg14 regulates autophagosome–lysosome fusion through Stx17 binding [27,28]. Therefore, we studied the autophagic degradation-suppressive effect of CbpC in 293T cells, employing p62 as a reporter for autophagic flux. When CbpC was transiently expressed in 293T cells, acute autophagy induction via Atg14–CbpC–p62 signaling led to p62 degradation (Fig EV3F). Therefore, we constructed MEF cells stably expressing GFP-CbpC$_{T4}$ or GFP to determine the autophagic degradation-suppressive effect of CbpC. Upon autophagy activation induced by rapamycin treatment, p62 was degraded in control MEFs stably expressing GFP (Fig 3L). In contrast, p62 degradation following rapamycin treatment was dramatically suppressed in MEFs stably expressing GFP-CbpC$_{T4}$, which also supported our notion that CbpC inhibits autophagosome–lysosome fusion and subsequent autophagic degradation. We therefore addressed whether CbpC-induced Atg14 depletion could manipulate Atg14–Stx17 interactions. Thus, we measured direct interactions between these proteins in the presence or absence of CbpC in living cells using nano-bioluminescence resonance energy transfer (NanoBRET) [29]. Notably, the Atg14–Stx17 interaction was robustly suppressed in the presence of CbpC (Fig 3M); however, only slight inhibition of the Atg14–Beclin1 interaction occurred and CbpC–Atg14–Beclin1 complex formation was observed (Fig EV3G and H). Together, these results supported our notion that *S. pneumoniae* can manipulate autophagy by employing CbpC as a decoy to cause autophagic degradation of Atg14 and subsequent suppression of autophagosome–lysosome fusion and bactericidal autophagic degradation (Fig EV3I).

### Interaction of the dp3 loop of CbpC with the coiled-coil domain (CCD) of Atg14

To identify the region of Atg14 responsible for CbpC binding, we prepared a series of GFP–Atg14 truncation mutants, including the N

(residues 1–70), CCD, and ΔCCD variants, and the full-length (FL) control Atg14 protein (Fig 4A). We then determined their abilities to interact with GST-CbpF$_{R6}$ by performing GST-pulldown assays. The Atg14 CCD, which Atg14 heterodimerizes with the Beclin1 CCD [30], bound strongly to GST-CbpF$_{R6}$ (Fig 4B). In contrast, UVRAG CCD, which also heterodimerizes with Beclin1 through its CCD, did not bind to GST-CbpF$_{R6}$ (Fig EV4A). We then analyzed truncation variants of CbpF$_{R6}$ to identify regions responsible for binding to the Atg14 CCD. Based on the structure of CbpF$_{R6}$ [10], we designed a series of GST-CbpF$_{R6}$ truncation mutants, i.e., N, C, N1, N2, N1-2, dp3, dp4, and an in-frame deletion mutant devoid of the entire dp3 region (Δloop) (Fig 4C). We then determined their binding activities to GFP-Atg14 CCD in GST-pulldown assays. We found that the loop structure in the dp3 domain in the N-terminal region of CbpF$_{R6}$ was pivotal for Atg14 CCD binding (Fig 4D and E).

Based on the structure of CbpF$_{R6}$ [10], we focused on two protruding loops in the dp3 domain (Fig 4F). We prepared two CbpF$_{R6}$ mutants with single-residue substitutions (Y83A and E95A; Fig 4F and G). Subsequent GST-pulldown assays using these variants and lysates from 293T cells expressing GFP-Atg14 CCD revealed that Y83 in the loop was essential for Atg14 CCD binding since the Y83A mutant (but not the E95A mutant) failed to interact with Atg14 (Fig 4H). Consistently, the accumulation of LC3-II and intrinsic puncta formation decreased robustly in GFP-CbpC$_{T4}$ Y83A-expressing cells, when compared with cells expressing E95A or WT GFP-CbpC$_{T4}$ (Fig 4I and J). These results suggest that the CbpC dp3–Atg14 CCD interaction promotes autophagy induction. We next addressed the possibility of CbpC Y83 phosphorylation by performing an IP experiment using GFP-CbpC$_{T4}$ FL, Y83A, and Δloop in the presence of phosphatase inhibitor; however, no phosphorylation signal was detected (Fig EV4B). Furthermore, we investigated whether the CbpC dp3–Atg14 interaction functions in Atg14 degradation by transiently expressing GFP-CbpC$_{T4}$ Δloop in 293T cells, and we found that Atg14 degradation was not affected by the GFP-CbpC$_{T4}$ Δloop construct, which is deficient in Atg14 binding (Fig EV4C). These results also strengthen our notion

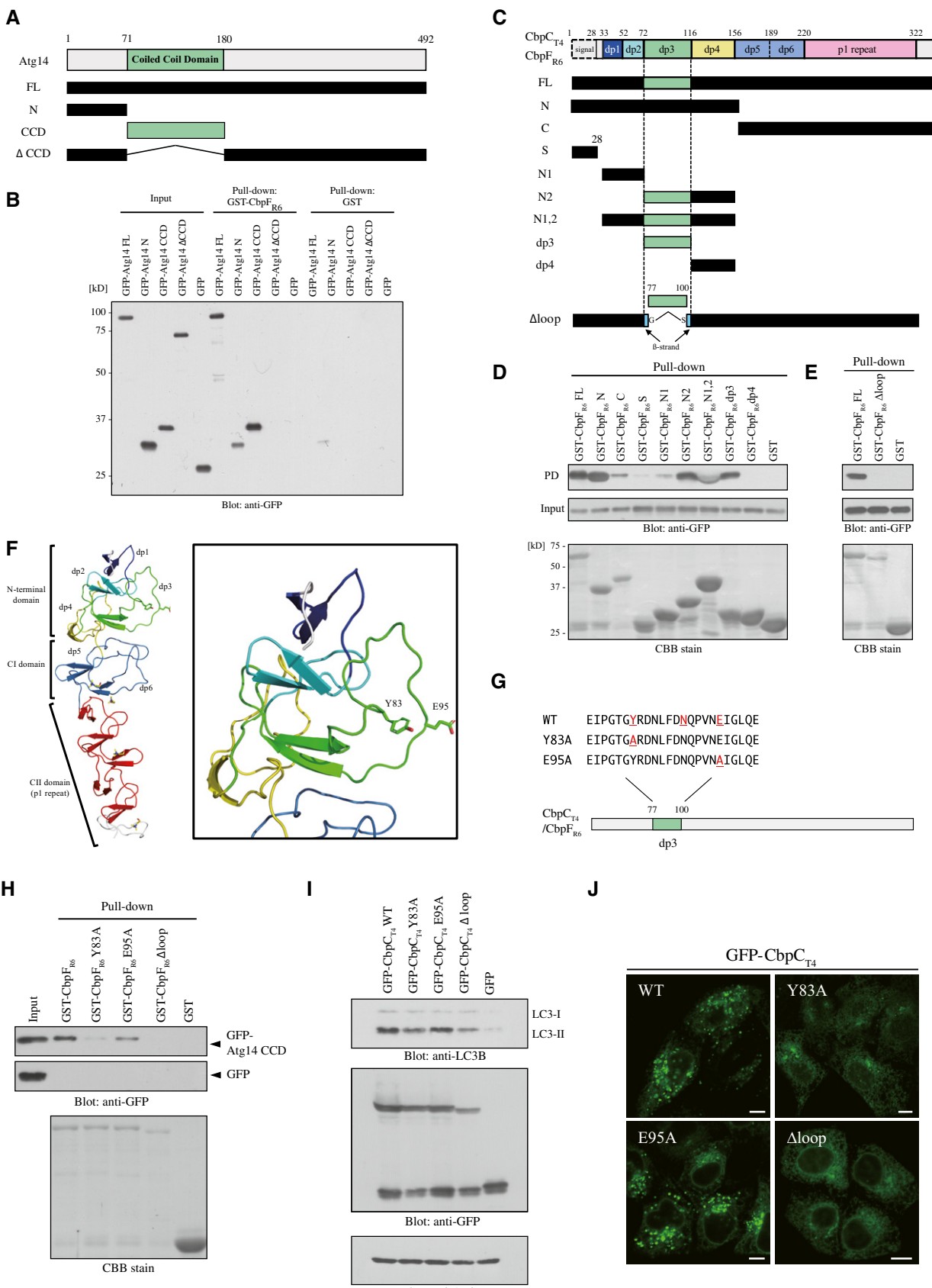

**Figure 4.**

**Figure 4.  The loop structure in the CbpC dp3 domain enabled interaction with the CCD of Atg14.**

A  Diagram of the Atg14 derivatives used in (B).

B  GST-pulldown assays performed using GST-CbpF$_{R6}$ or GST and lysates from 293T cells expressing GFP-Atg14 derivatives or GFP. The bound proteins were analyzed by immunoblotting using an anti-GFP antibody.

C  Schematic representation of the CbpF$_{R6}$ derivatives used in (D) and (E). The domains are color-coded according to the crystal structure report for CbpF$_{R6}$ (lower left) [10].

D, E  GST-pulldown assays using GST-CbpF$_{R6}$ derivatives and lysates from 293T cells expressing GFP-Atg14 CCD or GFP. The bound proteins were analyzed by immunoblotting using an anti-GFP antibody. Each GST-CbpF$_{R6}$ derivative-bound bead was confirmed by Coomassie brilliant blue (CBB) staining.

F  Structure of CbpF$_{R6}$ (PDB ID: 2V04). The overall structure (left) and dp3 domain (right) of homology-modeled structures are shown. The side chains of Y83 and E95 and bound choline molecules are shown as green and yellow sticks, respectively.

G  The amino acid sequences of the CbpC$_{T4}$/CbpF$_{R6}$ mutants used in (H), (I), and (J).

H  GST-pulldown assays performed using GST-CbpF$_{R6}$ mutants or GST and lysates from 293T cells expressing GFP-Atg14 CCD or GFP. The bound proteins were analyzed by immunoblotting using an anti-GFP antibody. Each GST-CbpF$_{R6}$ mutant-bound bead was confirmed by CBB staining.

I  Lysates of 293T cells transiently expressing GFP-CbpC$_{T4}$ mutants or GFP were subjected to SDS–PAGE and analyzed by immunoblotting using antibodies against LC3, GFP, or actin.

J  Confocal microscopy images of HeLa cells transiently expressing the indicated GFP-CbpC$_{T4}$ mutants. Scale bars, 10 μm.

Source data are available online for this figure.

that selective autophagy mediated by the Atg14–CbpC–p62 axis was involved in Atg14 degradation.

## Domain analysis of CbpC and the involvement of p62 in their interaction

We performed a domain analysis of CbpC in terms of p62 binding by preparing truncated versions of CbpC$_{T4}$ (Fig 5A). We found that the dp5 loop of CbpC was responsible for p62 binding (Fig 5A and B). Subsequent IP assays revealed that residues D167–D168 in the dp5 loop were required for the CbpC–p62 interaction (Fig 5C–E). Therefore, we examined whether the CbpC–p62 interaction was involved in autophagy activation by employing truncated versions of CbpC, which were deficient in p62-binding (Fig EV5A). Our data revealed that the CbpC–p62 interaction was required for intrinsic puncta formation by CbpC and autophagy activation (Fig EV5B and C). Intriguingly, intracellular localization of CbpC in the vicinity of the endoplasmic reticulum was observed only with Atg14-interacting CbpC variants (Fig EV5B). Together, these results also support our notion that Atg14–CbpC–p62 complex formation is required for autophagy activation.

We next performed a domain analysis of p62 in terms of CbpC binding (Fig 5F). We found that the CbpC-binding capacity decreased dramatically with the TRAF6-binding-deficient mutant, but only decreased partially with the Keap1-binding-deficient mutant (Fig 5G). These results suggested the possible involvement of TRAF6 in the CbpC–p62 interaction. We therefore examined whether the CbpC–p62 interaction was facilitated by the presence of TRAF6. IP assays using 293T cells co-expressing GFP-CbpC$_{T4}$ and p62-3Myc in the presence or absence of FLAG-TRAF6 revealed that the CbpC–p62 interaction was robustly facilitated by the presence of TRAF6, regardless of its E3 ligase activity (Fig EV5D). However, no direct interaction of TRAF6 with the Atg14–CbpC–p62 complex was detected (Fig EV5E). Furthermore, the CbpC–p62 interaction robustly decreased with CbpC$_{T4}$ Δdp5, even in the presence of TRAF6 (Fig EV5F). To investigate the physiological role of the CbpC–p62 interaction via CbpC dp5, we constructed *S. pneumoniae* R6 Δ*cbpF* derivatives complemented with a series of p62-interaction-deficient CbpF$_{R6}$ variants via homologous recombination, and we conducted Atg14-degradation assays using A549 cells. Notably, we found that the disappearance of perinuclear Atg14 caused by

*S. pneumoniae* infection was dramatically attenuated in *S. pneumoniae* strains expressing p62-binding-deficient CbpF$_{R6}$ (Figs 5H and I, and EV5G).

## Autophagy induction activity via Atg14 binding was a distinctive property of CbpC$_{T4}$ and CbpF$_{R6}$ from their orthologs

Basic Local Alignment Search Tool (BLAST) analysis suggested that CbpC$_{T4}$ orthologs exist in multiple *Streptococcus* strains, including *S. mitis* and *S. oralis* (Fig 6A). Furthermore, the *S. pneumoniae* TIGR4 strain has two CbpC$_{T4}$ orthologs (CbpJ and CbpF$_{T4}$), and the R6 strain also has two CbpC$_{T4}$-orthologs (CbpF$_{R6}$ and PcpC), as shown in Figs 6B and EV2E [10]. As the amino acid sequences of dp3 in these homologs were distinct, we classified them into three groups (Fig 6B). The crystal structure of CbpJ [11] and a homology model of PcpC were compared with the structure of CbpF$_{R6}$ (Fig 6C). While PcpC lacks the N-terminal portion of the CBR (Fig EV2E), homology modeling indicated that both CbpJ in TIGR4 and PcpC in R6 have retained the basic structure of the dp3 domain. However, the loops of CbpJ and PcpC, which contain residues corresponding to Y83 (S85 in CbpJ and V85 in PcpC), have distinct structures from that of CbpF$_{R6}$.

We then examined the binding capacity of CbpJ from TIGR4 (Group 2, Fig 6B) and PcpC from R6 (Group 3, Fig 6B) to Atg14 CCD by performing GST-pulldown assays. We found that CbpJ and PcpC were unable to bind Atg14 (Figs 6D and EV5H). The increase in LC3-II protein level and puncta formation was not observed in GFP-CbpJ- or GFP-PcpC-expressing cells (Fig 6E and F), showing that autophagy induction by Atg14 CCD binding was an intrinsic property of CbpC$_{T4}$ (Group 1).

We next prepared dp3 domain-swapped proteins (Figs 6G and EV5H) and measured their binding to the Atg14 CCD in GST-pulldown assays. Interestingly, the Atg14-binding capacity was abolished with CbpF$_{R6, dp3J}$, where dp3 from CbpF$_{R6}$ was swapped with dp3 from CbpJ (Fig 6H). The increase in LC3-II protein level and intrinsic puncta formation was also abolished in cells expressing CbpC$_{dp3J}$, where dp3 from CbpC$_{T4}$ was swapped with dp3 from CbpJ (Fig 6E and F). These results also suggest that the CbpC–Atg14 CCD interaction was required for autophagy induction. The Atg14 CCD-binding abilities of CbpJ$_{dp3C}$ (dp3 in CbpJ swapped with dp3 from CbpC$_{T4}$) and PcpC$_{dp3C}$ (dp3 in PcpC was swapped with dp3 from

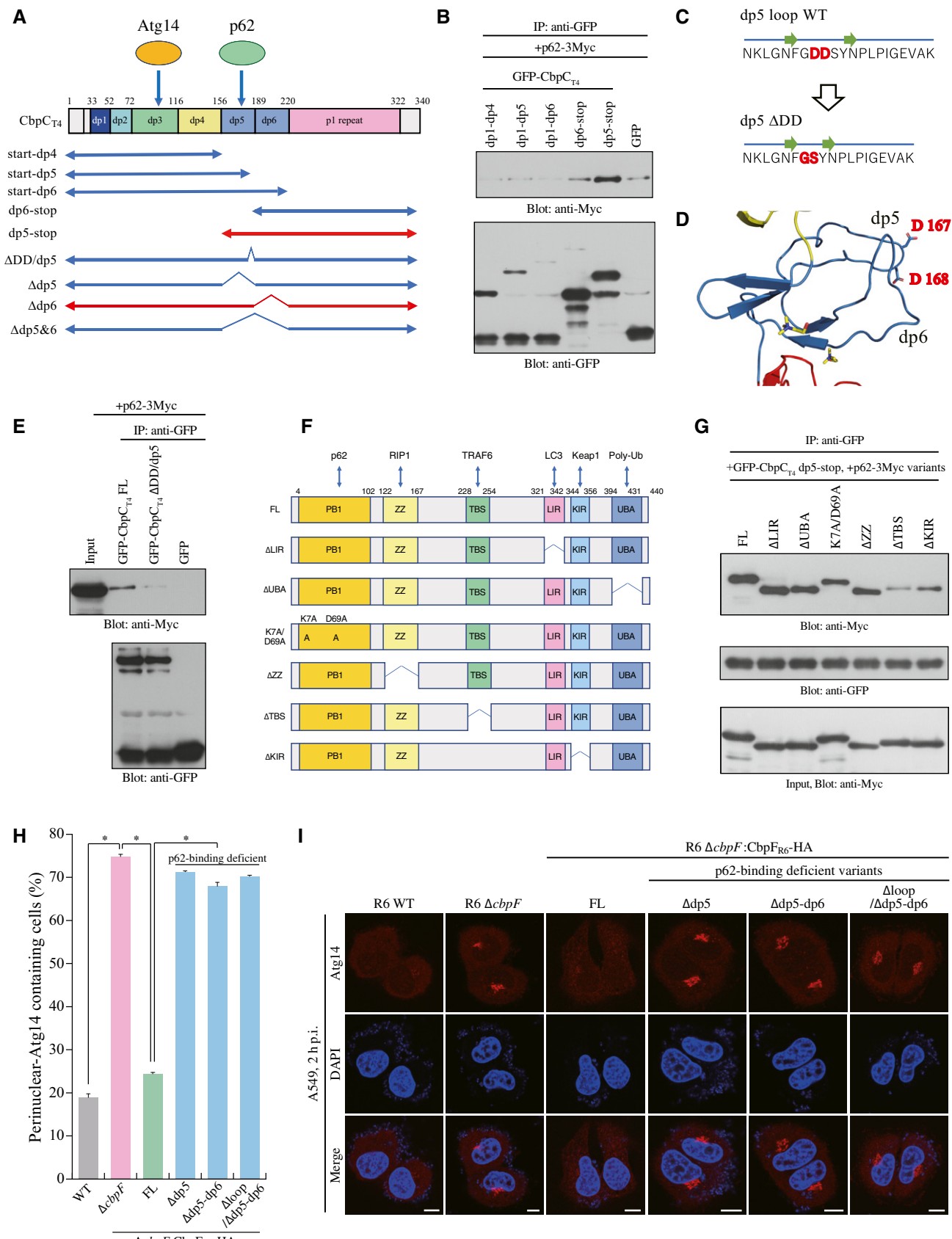

**Figure 5.**

**Figure 5. Domain analysis of the CbpC–p62 interaction.**

A Schematic representation of the CbpC$_{T4}$ variants used in (B), (E), (H), and (I).

B Lysates from 293T cells transiently expressing p62-3Myc and GFP-CbpC$_{T4}$ variants were immunoprecipitated using a GST-GFP-Nanobody fusion protein. Bound proteins were analyzed by immunoblotting.

C Diagram of the ΔDD mutation in dp5 loop domain.

D Structures of the dp5 and dp6 domains in CbpF$_{R6}$. The side chains of D167–D168 are shown.

E Lysates from 293T cells transiently expressing p62-3Myc and GFP-CbpC$_{T4}$ variants were immunoprecipitated using the GST-GFP-Nanobody protein. Bound proteins were analyzed by immunoblotting.

F Diagram of the p62 variants used in (G).

G Lysates of 293T cells transiently expressing GFP-CbpC$_{T4}$ and p62-3Myc variants were immunoprecipitated with the GST-GFP-Nanobody protein. Bound proteins were analyzed by immunoblotting using the indicated antibodies.

H A549 cells infected with the indicated *S. pneumoniae* strains for 2 h were fixed and stained with DAPI and an anti-Atg14 antibody, and the percentages of perinuclear-localizing Atg14 containing cells were quantified.

I Representative epifluorescence images in (H) are shown. Scale bars, 10 μm.

Data information: In (H), data represent mean ± SEM of 3 biological replicates. Student's *t*-test was used to calculate statistical significance. *$P < 0.01$.

Source data are available online for this figure.

CbpC$_{T4}$) were partially restored (Figs 6I and EV5I). Together, these results strongly suggest that autophagy induction through Atg14 binding was an intrinsic property of the Group 1-CbpC$_{T4}$ family.

## Discussion

In this study, we identified CbpC from the *S. pneumoniae* TIGR4 strain as an autophagy-activating virulence factor. We found that CbpC binding to Atg14 was required for autophagy activation. Our data further revealed that selective autophagy triggered by the Atg14–CbpC–p62 axis promoted the autophagic degradation of Atg14, resulting in suppressed autophagosome–lysosome fusion, which contributes to the intracellular viability of *S. pneumoniae*.

To understand these phenomena in more detail, we demonstrated that dp3 in the N-terminal region of CbpC was responsible for its binding to the CCD of Atg14. Further domain analysis of Atg14 binding to CbpC revealed that the Y83 residue in dp3 was crucial for Atg14 CCD binding. Indeed, the Y83A and dp3 loop-deleted CbpC mutants were deficient in autophagy induction. The N-terminal region is also involved in binding to LytC, a muramidase involved in pneumococcal autolysis [10]. LytC acts as a lysozyme during growth at 30°C, and the CbpF$_{R6}$–LytC interaction suppresses autolysis [10]. It is not yet known whether the binding of LytC and CbpC$_{T4}$ to the N-terminal region of Atg14 is competitive. Our findings also revealed that CbpC can interact with p62 via the dp5 domain. The CbpC–p62 interaction was dramatically promoted by TRAF6. CbpC showed no binding capacity for TRAF6. The molecular mechanism whereby TRAF6 promotes the CbpC–p62 interaction remains to be elucidated.

The *S. pneumoniae* TIGR4 strain has two CbpC$_{T4}$ orthologs, namely CbpJ and CbpF$_{T4}$; the R6 strain also has two CbpC$_{T4}$-orthologs, namely CbpF$_{R6}$ and PcpC (Figs 6B and EV2E) [10]. The coexistence of these two highly similar orthologs in the TIGR4 and R6 strains indicates the functional correlation of these genes. These pneumococcal proteins have the same architecture as CbpC$_{T4}$ and are likely to constitute a CbpC$_{T4}$-like subfamily. They each have a typical CBM in their C-terminal domain and a similar structural framework in their N-terminal domain (composed of several non-consensus CBRs), as well as distinctive insertion sequences. These intrinsic insertions between two β-strands form connecting loops

and might provide each CbpC$_{T4}$-like protein with an intrinsic physiological function. Here, we showed that dp3 and dp5 from CbpC can bind to Atg14 and p62, respectively; however, the Atg14-binding capacity was not seen in dp3 from CbpJ and PcpC. Although the binding partner of dp3 and dp5 from CbpJ and PcpC remains unknown, it is can be speculated that the CbpC$_{T4}$-like subfamily might serve a similar intracellular function as that of CbpC$_{T4}$.

In mammalian cells, Atg14 levels are controlled by ZBTB16 E3 ligase via G-protein-coupled receptor (GPCR) stimulation [31]. Upon GPCR stimulation, ZBTB16 promotes Atg14 degradation to inhibit autophagy. Here, we demonstrated a sophisticated pathogen-driven Atg14-degradation mechanism. Intriguingly, *S. pneumoniae* utilized host-selective autophagy to reduce Atg14, resulting in the suppression of continuous autophagic activation and autophagosome–lysosome fusion. Atg14 has two functions in autophagy. During the initiation of autophagy, Atg14 acts as an autophagy-specific regulator of the class III phosphatidylinositol 3-kinase complex (PI3KC3) to generate PI3P [24]. During autophagy maturation, Atg14 promotes STX17–SNAP29–VAMP8-mediated autophagosome fusion with lysosomes [27]. In this study, we demonstrated that CbpC has two distinct effects on autophagy: It activates autophagy initiation and suppresses autophagosome–lysosome fusion. Ectopic CbpC expression increased autophagy flux and elevated LC3-II levels, even in the presence of chloroquine. However, it also caused subsequent Atg14 degradation and attenuated autophagic degradation. This duality is reminiscent of the effects of the VacA toxin of *Helicobacter pylori* on autophagic activation [32]. A short period of VacA treatment (6 h) triggers autophagy; however, more prolonged VacA treatment (24 h), similar to the conditions observed during chronic infection, can inhibit autophagic degradation by disrupting autophagic flux. We demonstrated that CbpC inhibited autophagic flux; however, a detailed time course of the changes of CbpC-driven autophagic activation and subsequent Atg14 depletion during pneumococcal infection has not been fully revealed. Furthermore, prolonged Atg14 depletion can prevent not only on autophagosome–lysosome fusion, but also continuous autophagic induction. In this study, we did not clarify whether CbpC$_{T4}$-driven Atg14 depletion suppressed both autophagy activation and autophagosome–lysosome fusion. It is captivating to speculate that intracellular *S. pneumoniae* can establish prolonged or chronic infection by deploying CbpC to subvert both autophagic induction and degradation.

PI3K3C3 forms two mutually exclusive complexes that execute distinct functions [25,26]. One complex is with Atg14 and promotes autophagy initiation, whereas the other complex replaces Atg14 with UVRAG–Rubicon and suppresses autophagy. Considering that CbpC cannot bind to the CCD domain of UVRAG (Fig EV4A), we conclude that CbpC-triggered autophagy is unlikely due to inhibited

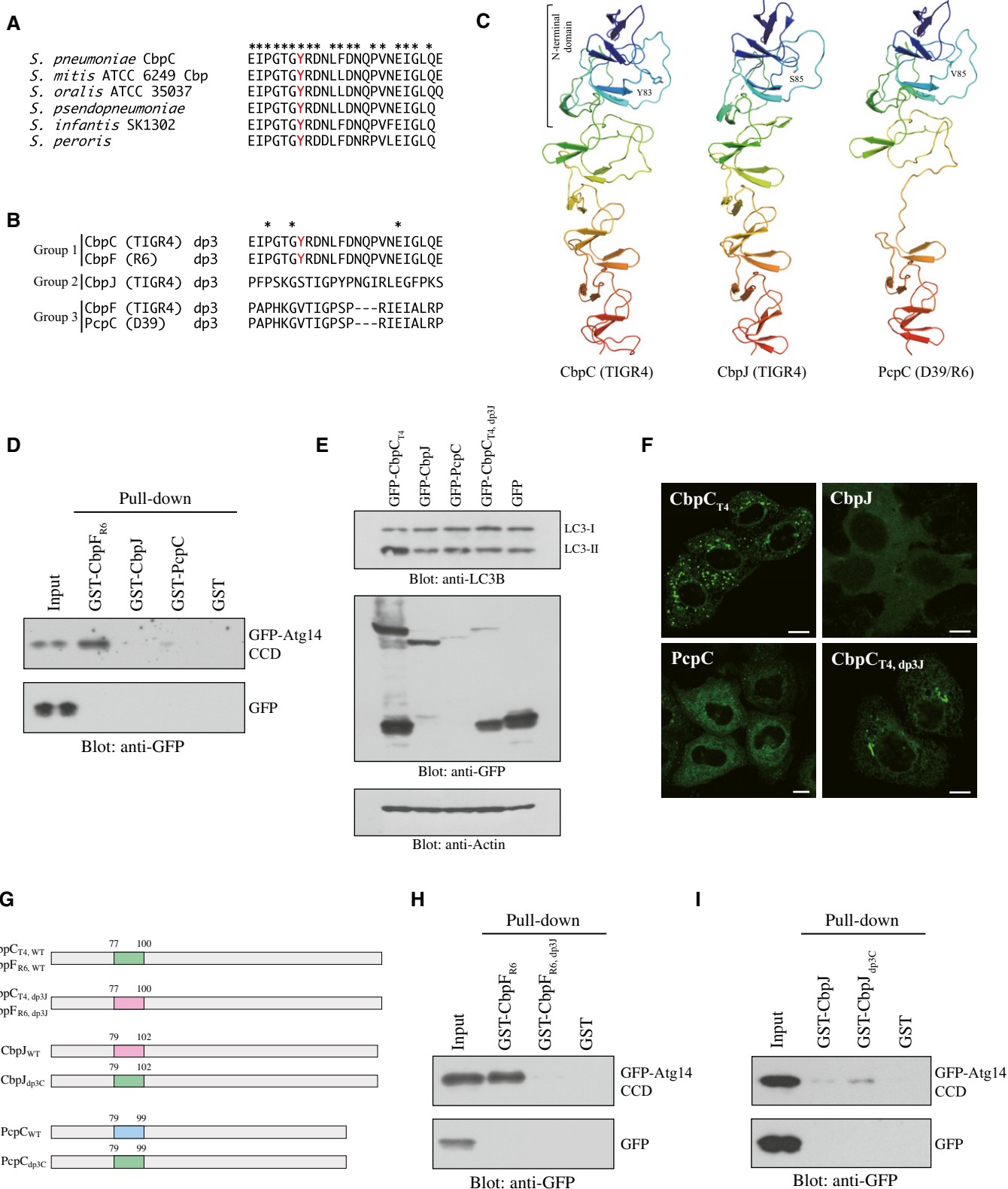

**Figure 6.**

**Figure 6. Comparison of the Atg14-binding capacities and autophagy induction activities of CbpC family proteins.**

A Comparison of the amino acid sequences of the dp3 domains from CbpC$_{T4}$ orthologs of other *Streptococcus* strains. The identical amino acid residues in all sequences in (A) and (B) are indicated by asterisk.

B Comparison of the amino acid sequences of the dp3 domains in CbpC$_{T4}$ orthologs CbpJ, CbpF$_{T4}$, CbpF$_{R6}$, and PcpC.

C Full-length structures of CbpC$_{T4}$ (PDB ID: 2V04), CbpJ (PDB ID: 6JYX), and PcpC (homology model).

D GST-pulldown assays using the indicated GST-fusion proteins or GST and lysates from 293T cells expressing GFP-Atg14 CCD or GFP. Bound proteins were analyzed by immunoblotting using an anti-GFP antibody.

E Lysates from 293T cells expressing GFP-fusion proteins or GFP were subjected to SDS–PAGE and analyzed by immunoblotting using antibodies against LC3, GFP, or actin.

F Confocal images of HeLa cells transiently expressing GFP-fusion proteins. Scale bars, 10 μm.

G Diagram of CbpC$_{T4}$/CbpF$_{R6}$, CbpJ, and PcpC dp3 domain swaps with CbpJ or CbpC$_{T4}$/CbpF$_{R6}$.

H, I GST-pulldown assays using the indicated GST-fused proteins or GST and lysates from 293T expressing GFP-Atg14 CCD or GFP. Bound proteins were analyzed by immunoblotting using an anti-GFP antibody.

Source data are available online for this figure.

UVRAG–Beclin1 complex formation or promotion of the Atg14–Beclin1 interaction.

Many pathogens suppress autophagosome–lysosome fusion by targeting SNARE proteins, such as SNAP29 and Stx17. The 3C protease from enterovirus D68 cleaves the autophagic SNARE protein, SNAP29, to inhibit autophagic flux and to promote viral replication [33]. The 3C protease from coxsackievirus B3 cleaves SNAP29 and PLEKHM1 to facilitate its own propagation [34]. Human parainfluenza virus, type 3 deploys a phosphoprotein to inhibit autophagic flux [35]. The phosphoprotein interferes with the SNAP29–Stx17 interaction, resulting in the prevention of autophagosome–lysosome fusion. Lpg1137 from *Legionella pneumophila*, an effector protein acting as a serine protease, can cleave Stx17 during *Legionella* infection to inhibit autophagic flux [17]. In this study, we revealed novel bacterial tactics for targeting Atg14 to dampen autophagic degradation. *S. pneumoniae* hijacks host autophagy to deplete Atg14 by deploying CbpC.

In summary, we showed that *S. pneumoniae* manipulates the selective autophagy system to spatiotemporally regulate the level of Atg14 and dampening antibacterial xenophagy induced at later stages of infection, thereby increasing its viability and ability to invade more deeply into host tissues. Our discovery provides insight into the battles between intracellular pulmonary bacterial pathogens and host-defense systems; it may facilitate the development of therapeutic targets for controlling pneumococcal diseases.

## Materials and Methods

### Bacterial strains

The *S. pneumoniae* strains R6 (ATCC BAA-255) and TIGR4 (ATCC BAA-334) were purchased from the American Type Culture Collection. *S. pneumoniae* were grown in standing cultures of Todd–Hewitt Broth (THY; Becton Dickinson [BD], San Jose, CA, USA) containing 0.5% yeast extract (BD) broth or were plated on THY agar plates or Columbia agar plates with 5% sheep blood (BD) at 37°C in 5% $CO_2$. The *E. coli* strains MC1061, DH10B, BL21, and C43 (Cosmo Bio, Tokyo, Japan) were used for DNA cloning and protein expression, and were grown in LB medium or on LB-agar plates supplemented with 100 μg/ml ampicillin or 50 μg/ml kanamycin.

### Reagents and antibodies

Antibodies against GFP (D5.1), phosphotyrosine (P-Tyr-100), and LC3A/B (D3U4C) (Cell Signaling Technology, Massachusetts, USA); Myc (A14) and β-actin antibodies (C4, Santa Cruz Biotechnology, Texas, USA); GM130 (35/GM130, BD); HA (TANA2), Atg14 (PD026), Atg16L1 (M150-3), p62 (PM045), and poly-Ub (FK2) (MBL, Nagoya, Japan); and FLAG (FUJIFILM Wako Pure Chemical, Osaka, Japan) were used as primary antibodies. Horseradish peroxidase-conjugated goat anti-rabbit or anti-mouse antibodies (The Jackson Laboratory, Maine, USA) were used as secondary antibodies in immunoblotting experiments. Anti-Myc (9B11, Cell Signaling Technology), Protein G PLUS-Agarose (sc-2002, Santa Cruz Biotechnology), EZview Red Anti-HA Affinity Gel (Sigma, MO, USA), and GFP-Trap (Chromotech, Planegg-Martinsried, Germany) were used for IP experiments. Autophagy was induced and inhibited by 10 μM rapamycin, 40 μM chloroquine (Selleck Chemical, Texas, USA), and 100 nM Bafilomycin A1 (AdipoGen, CA, USA). Cycloheximide (50 μg/ml, FUJIFILM Wako Pure Chemical) was used to inhibit protein synthesis. All other reagents were purchased from FUJIFILM Wako Pure Chemical.

### Plasmids

Polymerase chain reaction (PCR) experiments were performed with Q5 High-Fidelity DNA Polymerase (New England BioLabs, Massachusetts, USA), and cDNA for reverse transcriptase-PCR (RT–PCR) experiments was synthesized using the SuperScript III One-Step RT-PCR System with Platinum Taq (Thermo Fisher Scientific, Massachusetts, USA). To generate the pEGFP-Cbp constructs, the *cbp* and *lytR* genes of *S. pneumoniae* TIGR4 and R6, and the *pspC* gene of *S. pneumoniae* R6 were subcloned into pEGFP-C1 (Takara Bio, Shiga, Japan) and pGEX6P-1 (GE Healthcare, Illinois, USA). Rat LC3B cDNA was subcloned into pmCherry-C1 (Takara Bio). Human Atg14 and TBK1 cDNAs were subcloned into pEGFP-C1. Human EGFR cDNA was subcloned into pEGFP-N3. Myc-tagged human Atg14 cDNA was subcloned into pcDNA3.1 (Thermo Fisher Scientific) for expression in mammalian cells or pTB101 [36] for expression in *E. coli* cells. Hemagglutinin (HA)-tagged human Atg14 cDNA was subcloned into pMXs-puro (Cosmo Bio) or pcDNA6.2 (Thermo Fisher Scientific). GFP and GFP-CbpC cDNAs were subcloned into pMXs-puro or pMXs-blast. Mammalian expression vectors for pIgR, Atg5-3Myc, p62-3Myc, GFP-Atg5, GFP-Atg12, GFP-Atg16L1, GFP-

Beclin1, GFP-WIPI1, and GFP-WIPI2 were constructed previously [18,37]. Human Atg14 cDNA was subcloned into pHTN HaloTag and pNLF1-N (Promega, Wisconsin, USA), human Stx17 cDNA was subcloned into pHTN HaloTag, and human Beclin1 cDNA was subcloned into pNLF1-N. Plasmids encoding Atg14 ΔCCD, CbpC (Δloop, Y83A, E95A), and swapped CbpC, CbpJ, and PcpC derivatives were designed using NEBuilder (New England BioLabs). The GST-GFP-Nanobody expression vector [38] was a generous gift from Dr. Yohei Katoh (Kyoto University). The sequences of the primers used in this study are shown in Appendix Table S1.

## Cell culture and transfection

The HeLa (human cervical cancer), 293T and 293A (human embryonic kidney fibroblasts), A549 (human lung epithelial cells), and MEF (mouse embryonic fibroblasts) cell lines [39,40] were cultured in Dulbecco's modified Eagle's medium (DMEM, Nakalai Tesque, Kyoto, Japan) supplemented with 10% fetal calf serum (FCS; Gibco-Thermo Fisher Scientific, Massachusetts, USA), 100 µg/ml gentamicin (FUJIFILM Wako Pure Chemical), and 60 µg/ml kanamycin (FUJIFILM Wako Pure Chemical). For retroviral production, platE cells were maintained in DMEM with 10% FCS, 1 µg/ml puromycin (Sigma), and 10 µg/ml blasticidin (Kaken Pharmaceutical, Tokyo, Japan). MEF-derived stable clones were cultured in DMEM containing 10% FCS, 1 µg/ml puromycin, and/or 10 µg/ml blasticidin. Transfections were performed with PEI MAX (Polysciences, Pennsylvania, USA), Lipofectamine LTX (Thermo Fisher Scientific), or the Effectene Transfection Reagent (Qiagen, Rhine-Westphalia, Germany), according to the manufacturers' protocols.

## Fluorescence microscopy

HeLa cells expressing GFP-Cbps were fixed with 4% paraformaldehyde in phosphate buffer (FUJIFILM Wako Pure Chemical) for 15 min at room temperature and washed three times with phosphate-buffered saline (PBS). After washing with distilled water, the specimens were mounted with Vector Shield (Vector Laboratories, CA, USA) and analyzed by confocal microscopy (LSM700, Zeiss, Baden-Württemberg, Germany).

## Recombinant retroviruses and infections

Recombinant retroviruses were prepared as previously described [19,20]. Briefly, retroviral plasmids were transfected into platE cells for 2 days, after which time the culture supernatant containing retrovirus was collected and centrifugated. Cleared supernatant was used for the retroviral infections. For retroviral infection, recipient cells were infected with retroviruses by adding cleared supernatant in the presence of polybrene for 6 h. Stable transformants were selected in DMEM/10% FCS with 1 µg/ml puromycin or 10 µg/ml blasticidin.

## Small interfering RNA (siRNA) experiments

siRNAs were synthesized and duplexed by siRNA Co., Ltd (Tokyo, Japan). The sequences of siRNAs targeting human pIgR, Atg5, or p62 mRNA are shown in Appendix Table S2. The siRNAs were reverse transfected into cells using Lipofectamine RNAi MAX (Thermo Fisher Scientific), according to the manufacturer's protocol. The knockdown efficiency was checked by RT–PCR using the primer pairs shown in Appendix Table S2.

## Construction of the *S. pneumoniae* mutants and complemented strains

Target gene inactivation in *S. pneumoniae* strain TIGR4 or R6 was performed as described previously [20]. Briefly, an *erm* cassette with long flanking regions homologous to the target gene was generated using two-step PCR as described [41], using the primers listed in Appendix Table S3. The PCR products were introduced into competent *S. pneumoniae* TIGR4 or R6 cells as described previously [20], and the transformants were selected in 1 µg/ml erythromycin. Substitution of the target gene was confirmed by PCR with primers shown in Appendix Table S3. The *S. pneumoniae* Δ*ply* strain was previously reported [20]. To construct the *S. pneumoniae* R6 Δ*cbpF*/pCbpF-FLAG strain, a PCR product encoding the cbpF$_{R6}$-FLAG gene and its promoter was generated by PCR using the primers shown in Appendix Table S4. The PCR product was inserted into pMX1 [42], and the ligated plasmid was introduced into competent *S. pneumoniae* R6 Δ*cbpF* cells, as described previously [20], and the transformants were selected in 50 µg/ml spectinomycin. Target gene expression was confirmed by Western blotting. *S. pneumoniae* R6 Δ*cbpF* derivatives were complemented with a series of p62-interaction-deficient CbpF$_{R6}$ variants by homologous recombination. A *cat* cassette with cbpF$_{R6}$-HA variants and long flanking regions homologous to the target locus was generated using two-step PCR as described [41] with the primers shown in Appendix Table S4. The PCR products were introduced into competent cells of the *S. pneumoniae* strain R6 as described previously [20], and transformants were selected in 2.5 µg/ml chloramphenicol. Substitution of the target gene was confirmed by PCR with the primers shown in Appendix Table S4. The primers were designed to target cbpF$_{R6}$-HA variants into the cps2H gene (which is not expressed in the R6 strain) in reverse orientation relative to the *cps* operon.

## *In vitro* protein production and binding assays

Wheat germ cell-free protein production was performed as described previously [43]. Briefly, cDNA templates encoding FLAG or biotin ligase site (bls) epitopes were generated by PCR with split-primer sets for autophagy-related genes. Amplified cDNAs were then used to produce proteins by the bilayer method using the WEPRO1240 Expression Kit (CellFree Sciences, Ehime, Japan). Biotinylated CbpC or control GFP proteins were mixed with FLAG-tagged autophagy-related proteins and incubated for 2 h at 26°C in different wells of a 96-well plate. Next, 20 µl of pre-buffered Streptavidin MagneSphere Paramagnetic Particles (Promega) was added to each well and incubated for 30 min at room temperature, and then washed three times with 120 µl of PBS/0.1% NP-40. The beads were then suspended in 20 µl of 2× SDS sample buffer and boiled for 5 min for sodium dodecyl sulfate–polyacrylamide gel electrophoresis (SDS–PAGE) and immunoblotting analysis.

                                    

## GST-GFP-Nanobody protein purification

*Escherichia coli* BL21 cells harboring the pGST-GFP-Nanobody plasmid were cultured in LB medium with 50 μg/ml ampicillin for 2 h at 37°C. Isopropyl-1-thio-β-D-galactopyranoside (IPTG) was added to the culture medium at a final concentration of 0.1 mM. After incubation for 2 h at 37°C, the bacteria were harvested. To express GST-Cbps, *E. coli* cells were cultured at 20°C for 9 h, after which 0.1 mM IPTG was added and the bacteria were incubated at 20°C for an additional 18 h. Following IPTG induction, the bacteria were harvested by centrifugation (4,300 $g$, 10 min, 4°C) and suspended in wash buffer (Tris-buffered saline [TBS] + 1% Triton X-100 + 10 mM β-mercaptoethanol) containing ethylenediaminetetraacetic acid (EDTA)-free Complete protease-inhibitor cocktail (Sigma). After sonication (3 × 30 s on ice) and centrifugation (4,300 $g$, 10 min, 4°C), the cleared lysates were mixed with prebuffered glutathione Sepharose 4B (GE Healthcare), and GST-fusion proteins were purified according to the manufacturer's protocol.

## GST-pulldown assays using cell lysates

293T cells were transfected with the indicated plasmids in 6-well plates, suspended in 500 μl of wash buffer (TBS/0.5% NP-40) containing EDTA-free Complete protease-inhibitor cocktail, sonicated on ice for 10 s, and centrifuged at 16,900 $g$ at 4°C for 10 min. Five microliters of GST-GFP-Nanobody bound to glutathione Sepharose 4B was mixed with cleared 293T cell lysates and incubated with rotation for 2 h at 4°C. The beads were washed three times with 1 ml of wash buffer and suspended in 30 μl of 2× SDS sample buffer and boiled for 7 min. The bound proteins were analyzed by immunoblotting.

## IP from cell lysates

Cells (293T) were seeded into 6-well plates and transfected with the indicated plasmids. After 18 h, the cells were suspended in 500 μl of wash buffer (TBS with 5 mM $MgCl_2$, 1 mM EDTA, and 0.05% NP-40) containing Complete protease-inhibitor cocktail, sonicated on ice for 10 s, and centrifuged at 16,900 × $g$ at 4°C for 10 min. Five microliters of GST-GFP-Nanobody-bound glutathione Sepharose 4B beads, GFP-Trap, or EZview Red Anti-HA Affinity Gel were mixed with the cleared lysates and rotated for 2 h at 4°C. In phosphorylation assays, IP was performed using GFP-Trap in the presence of a phosphatase inhibitor (PhosphoStop, Sigma). IPs with an anti-Myc antibody with Protein G PLUS-Agarose were performed according to the manufacturers' protocols using IP buffer (50 mM Tris-HCl pH 7.4, 150 mM NaCl, 1 mM EDTA, 10% glycerol, and Complete protease-inhibitor cocktail-EDTA). The beads were washed three times with 1 ml of wash buffer, suspended in 30 μl of 2× SDS sample buffer, and boiled for 7 min. The bound proteins were analyzed by immunoblotting.

## Infecting cells with *Streptococcus pneumoniae* for Western blotting

MEFs/HA-Atg14/pIgR were infected with *S. pneumoniae* as previously described [20,44,45]. Briefly, MEFs/HA-Atg14 cells (seeded at $2 \times 10^5$ cells/well in 6-well plates) were infected with fresh

*S. pneumoniae* at a multiplicity of infection of 100 and then centrifuged at 1,000 rpm for 5 min at room temperature. The cells were then incubated for 1 h at 37°C in 5% $CO_2$ and washed three times with Hanks' balanced salt solution (HBSS). DMEM containing 10% FCS, 200 μg/ml gentamicin, and 200 U/ml catalase was added to each well, and the cells were incubated for 30 min to kill extracellular bacteria. After changing the medium to DMEM with 10% FCS, 100 μg/ml gentamicin, and 200 U/ml catalase (incubation buffer), the cells were incubated for the indicated periods at 37°C in 5% $CO_2$. Unless otherwise stated, inhibitors were included in the incubation buffer to avoid affecting the bacterial invasion efficiency. At each time point, the cells were lysed with 100 μl of 2× SDS sample buffer, and the lysate was boiled for 7 min. After sonication (10 × 1 s), equal volumes of lysates were separated by SDS–PAGE and analyzed by Western blotting using the indicated antibodies.

## NanoBRET assays

Cells (293A) in 96-well plates were transfected with the NanoLuc and HaloTag expression vectors at a 1:100 ratio in the presence or absence of GFP-CbpC- or GFP-expression vectors. At 48 h post-transfection, NanoBRET activity was measured using the NanoBRET Nano-Glo Detection System (Promega), as previously described [29].

## Intracellular bacterial-survival assay

Intracellular bacterial-survival assays were performed as described previously [20]. Briefly, MEFs/pIgR seeded on 24-well plates were infected with WT or Δ*cbpC S. pneumoniae*, as described above. After centrifugation, the cells were incubated for 1 h at 37°C in 5% $CO_2$, washed twice with HBSS, and then incubated for 15 min with 500 μl of DMEM containing 10% FCS, 200 μg/ml gentamicin, and 200 U/ml catalase. To kill extracellular bacteria, the cells were incubated for another 15 min with DMEM containing 10% FCS, 200 μg/ml gentamicin, 200 U/ml catalase, and 10 μg/ml penicillin G (FUJI-FILM Wako Pure Chemical). After washing the cells three times with HBSS, they were incubated with DMEM containing 10% FCS and 200 U/ml catalase for the indicated periods at 37°C in 5% $CO_2$, after which they were lysed in PBS with 1.0% saponin (Sigma). The lysates were serially diluted with PBS containing 0.1% saponin onto THY agar plates, and the number of intracellular bacteria was expressed as the number of colony-forming units (CFUs).

## Homology modeling

Homology modeling of CbpC from *S. pneumoniae* TIGR4 (locus tag: SP_0377) was performed using the SWISS-MODEL server [46]. The crystal structure of the mature CbpF protein from *S. pneumoniae* R6 (locus tag: SPR0337), which has 92.3% amino acid-sequence identity with SP_0377 [10], was used as a template. The GMQE and QMEAN values were 0.99 and 1.06, respectively, suggesting that the homology modeling was reliable.

## Quantification and statistical analysis

Data are shown as the mean ± standard error of the mean (SEM). $P$ values were calculated using Student's $t$-test. If the standard

deviation values were significantly different ($F < 0.05$), then $P$ values were calculated using the Mann–Whitney $U$ test with Prism 6 software.

Expanded View for this article is available online.

## Acknowledgements

We thank Drs. Masaaki Komatsu, Noboru Mizushima, and Toru Yanagawa for providing reagents. We thank Dr. Daisuke Takamatsu (National Agriculture and Food Research Organization) for kindly supplying the *E. coli-S. pneumoniae* shuttle vector, pMX1. This work was supported by Grant-in-Aid for Scientific Research (C) awards 19K07568, 16K08800, and 25460555 to Mi.O. from the Ministry of Education, Culture, Sports, Science and Technology (MEXT). This work was also supported by grants from the Naito Foundation and the Uehara Foundation. We would like to thank Editage (www.editage.jp) for English language editing.

## Author contributions

MOg, AR, and MO supervised the overall project. MOg, SS, SM, SF, and AR designed the experiments. SS, MOg, SM, and MT performed the experiments. SS, MOg, SM, MT, TI, SF, AR, and MOg analyzed the data. MOg wrote the manuscript with input from all co-authors.

## Conflict of interest

The authors declare that they have no conflict of interest.

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
