## [Review Process File · EMBO Reports]

***Streptococcus pneumoniae* hijacks host autophagy by deploying CbpC as a decoy for Atg14 depletion**

Sayaka Shizukuishi, Michinaga Ogawa, Satoko Matsunaga, Mikado Tomokiyo, Tadayoshi Ikebe, Shinya Fushinobu, Akihide Ryo, Makoto Ohnishi

Review timeline:

Submission date:	16 September 2019
Editorial Decision:	16 October 2019
Revision received:	13 January 2020
Editorial Decision:	10 February 2020
Revision received:	28 February 2020
Accepted:	6 March 2020

Editor: Achim Breiling

Transaction Report:

1st Editorial Decision

16 October 2019

Thank you for the transfer of your research manuscript to EMBO reports. We have now received reports from the three referees that were asked to evaluate your study, which can be found at the end of this email.

As you will see, all referees think that the findings are of interest and merit publication in EMBO reports, but they also have several comments, concerns and suggestions, indicating that a major revision of the manuscript is necessary. As the reports are below, and I think all points need to be addressed, I will not detail them here. However, I think that the three major points of referee #1 and the first major point of referee #3 need particular attention and need to be addressed experimentally.

Given the constructive referee comments, we would like to invite you to revise your manuscript with the understanding that all referee concerns must be addressed in the revised manuscript and/or in a detailed point-by-point response. Acceptance of your manuscript will depend on a positive outcome of a second round of review. It is EMBO reports policy to allow a single round of revision only and acceptance or rejection of the manuscript will therefore depend on the completeness of your responses included in the next, final version of the manuscript.

Revised manuscripts should be submitted within three months of a request for revision; they will otherwise be treated as new submissions. Please contact me if a 3-months time frame is not sufficient so that we can discuss the revisions further.

When submitting your revised manuscript, please also carefully review the instructions that follow below. Failure to include requested items will delay the evaluation of your revision. When submitting your revised manuscript, we will require:

- 1) a .docx formatted version of the final manuscript text (including legends for main figures, EV figures and tables), but without the figures included. Please make sure that the changes are highlighted to be clearly visible. Figure legends should be compiled at the end of the manuscript text.

2) individual production quality figure files as .eps, .tif, .jpg (one file per figure), of main figures and EV figures. Please upload these as separate, individual files upon re-submission.

The Expanded View format, which will be displayed in the main HTML of the paper in a collapsible format, has replaced the Supplementary information. You can submit up to 5 images as Expanded View. Please follow the nomenclature Figure EV1, Figure EV2 etc. The figure legend for these should be included in the main manuscript document file in a section called Expanded View Figure Legends after the main Figure Legends section. Additional Supplementary material should be supplied as a single pdf labeled Appendix. The Appendix should have page numbers and needs to include a table of content on the first page (with page numbers) and legends for all content. Please follow the nomenclature Appendix Figure Sx, Appendix Table Sx etc. throughout the text, and also label the figures and tables according to this nomenclature.

For more details please refer to our guide to authors:

See also our guide for figure preparation:

http://wol-prod-cdn.literatumonline.com/pb-assets/embosite/EMBOPress_Figure_Guidelines_061115-1561436025777.pdf

4) a complete author checklist, which you can download from our author guidelines (<https://www.embopress.org/page/journal/14693178/authorguide>). Please insert page numbers in the checklist to indicate where the requested information can be found in the manuscript. The completed author checklist will also be part of the RPF.

Please also follow our guidelines for the use of living organisms, and the respective reporting guidelines: <http://www.embopress.org/page/journal/14693178/authorguide#livingorganisms>

5) We strongly encourage the publication of original source data with the aim of making primary data more accessible and transparent to the reader. The source data will be published in a separate source data file online along with the accepted manuscript and will be linked to the relevant figure. If you would like to use this opportunity, please submit the source data (for example scans of entire gels or blots, data points of graphs in an excel sheet, additional images, etc.) of your key experiments together with the revised manuscript. If you want to provide source data, please include size markers for scans of entire gels, label the scans with figure and panel number, and send one PDF file per figure.

6) Our journal encourages inclusion of *data citations in the reference list* to directly cite datasets that were re-used and obtained from public databases. Data citations in the article text are distinct from normal bibliographical citations and should directly link to the database records from which the data can be accessed. In the main text, data citations are formatted as follows: "Data ref: Smith et al, 2001" or "Data ref: NCBI Sequence Read Archive PRJNA342805, 2017". In the Reference list, data citations must be labeled with "[DATASET]". A data reference must provide the database name, accession number/identifiers and a resolvable link to the landing page from which the data can be accessed at the end of the reference. Further instructions are available at:

7) Regarding data quantification and statistics, can you please specify, where applicable, the number "n" for how many independent experiments (biological replicates) were performed, the bars and error bars (e.g. SEM, SD) and the test used to calculate p-values in the respective figure legends. Please provide statistical testing where applicable, and also add a paragraph detailing this to the methods section. See:

<http://www.embopress.org/page/journal/14693178/authorguide#statisticalanalysis>

8) Please add up to 5 key words to the title page.

I look forward to seeing a revised version of your manuscript when it is ready. Please let me know if you have questions or comments regarding the revision.

REFEREE REPORTS

Referee #1:

In this work Shizukuishi and colleagues look at the regulation and avoidance of the autophagy pathway by *S. pneumoniae*. Specifically, they look at the action of CBPs in the activation of macroautophagy and clearance of ATG14L. Mechanistically, they propose that CbpC (from strain TIGR4) binds ATG14L and p62 resulting in the targeting of ATG14L to the autophagosome. Overall this is an extremely well written manuscript that proposes a novel and interesting mechanism. The experiments mapping the interaction between ATG14L and CbpC were exceptionally good. However, the conclusions and model seem premature based on the experimental data provided. This manuscript would benefit from additional experiments under conditions that are closer to physiologic to ensure the data generated from overexpression act similarly in a infection.

Major Concerns:

1. Fig 1. In the initial screens testing Cbp and autophagy it is not clear if overexpressed CbpC in the mammalian system is going to accurately recapitulate the effect of a cell surface proteins during infection or if bacterial autolysis releases large levels of free CbpC, enough to maintain repression of ATG14L1 levels.

Additionally, while survival was analyzed (1F) two controls were missing that I would consider critical. 1) the addition of a survival assay in an autophagy deficient background should be done. If the loss of CbpC results in higher autophagic killing then the viability should be the same in an autophagy deficient background, and 2) the model proposed would infer that cells that were not infected should not have an increase in autophagy. Given the MOI and cell type used one would expect that less than 100% of cells should have been infected. Immunofluorescent staining of autophagy and bacterial markers should be performed. Confusingly, based on data from later figures (p62 clearance), it seems that the majority of cells must degrading ATG14L in response to SP, which would indicate an infection rate of ~80% or more. Therefore, it would be good to show that the autophagic response that is being measured throughout is due to the CbpC release in infected cells.

2. Fig. 2. The interaction of ATG14 and CbpC was never shown endogenously. ATG14 was not shown to localize to invading SP. The overexpression ATG14 may well have thrown off the stoichiometry of ATG14-beclin. Which calls into question if the CbpC would be able to compete for ATG14 binding to Beclin-1. Since only small amounts of ATG14 are free of Beclin-1 (Kim et al Cell 2013) it is unclear based on the data shown if this interaction would occur under physiologic conditions.

3. Fig. 3. The loss of ATG14 (esp as viewed by IF is profound). The only link between ATG14 loss and autophagy is in Fig. 3B. However, in the delta-CbpC mutant ATG14 levels at lower in WT control, then higher at 3 hours. Effectively we have one lane that shows CbpC dependence and one that does not. No controls showing other cellular responses were not affected and that similar bacterial load was added. The rescue of ATG14 clearance in an autophagy deficient line or through pharmacological inhibitors, but not through the additional of proteasomal inhibitors would show the dependence on autophagy for ATG14 degradation. The IF data is strong; however, as with other figures there is no evidence that ATG14 is stable in cells that were not infected as would be predicted by the model. Perhaps a lower MOI could be used leaving a polyclonal population linking bacterial internalization to ATG14 loss.

Minor concerns:

- Model Fig. 1G would be more informative if a distinction was made between autophagy and xenophagy (or it could be removed)
- ATG14L was the old nomenclature, ATG14 is the current name (capitalization for proteins, such as ATG14 should also follow standard conventions <https://www.jci.org/kiosk/publish/genestyle>)
- Great mapping in Figure 4. Would have been nice to solidify some of the key previous data (ATG14 loss) with this mutant, underscoring the importance of the interaction
- Ub ligase activity of TRAF6 could have been tested or discussed in figure 5 as a modulator of complex formation

Referee #2:

This is a very interesting manuscript. Clearly autophagy is a relatively new area of study for pathogens that invade so this work is timely. Recent work from this group indicated that when pneumococci enter the cytoplasm, pneumolysin participates in directing the bacteria towards the autophagy pathway for degradation. This new manuscript describes a second safe path through cells. The existence of this second path fits with decades of work indicating that pneumococci can traverse endothelial and epithelial barriers by passing safely through cells (no indication of barrier breakdown). Thus, it is an important mechanistic study that establishes some details about an important step in invasive disease. The role of CbpC is a surprise that adds further spice and intrigue to the mechanism. This paper demonstrates the biochemistry of the CbpC interaction with autophagy components in model systems

The experiments are clearly presented with excellent controls. Thus, the hypotheses are supported by excellent data in model systems. The only comment is that all of the systems used are not related to cells pneumococci encounter during disease. It remains to be seen how this mechanism plays out in cells that represent the actual sites of transmigration during infection. Could the authors add images of CbpC and autophagy components colocalizing on bacteria crossing endothelial or epithelial barriers?

Referee #3:

In the present report, Shizukuishi et al investigated the role of the choline binding protein CbpC from *S. pneumoniae* in the host cell autophagic response during infection. They showed that CbpC induced the formation of LC3 autophagosome-like vesicles in HeLa cells. Moreover, they found that CbpC interacts with seven Atg proteins, including Atg5 and Atg14L. In a series of experiments, the authors identified the region in CbpC that binds Atg14L and that the interaction of CbpC with Atg14L was required to induction of autophagy. Interestingly, the authors did not find an increase of LC3-II levels in GFP-CbpC- or GFP-PcpC-expressing 293T cells, suggesting that the induction of autophagy by Atg14L CCD binding is an intrinsic property of CbpCT4. Overall, the study is well performed and well written. There is no much known about the interactions between *S. pneumoniae* and autophagy and this study provides a major contribution to this topic. However, there are some major points that are unclear and some experiments needed to confirm the findings using the bacteria.

Major points:

The data with the expression of CbpC, including the dp3 and dp5 CbpCT4 mutants should be complemented with experiments showing this is valid for the infection with *S. pneumoniae*. Not sure if all of these mutants are available but at least some infections with *S. pneumoniae* and show targeting either to autophagosomes or lysosomes. This is important to confirm the interactions reported here are also valid during infection of cells by *S. pneumoniae*.

In this context, Fig.3 should also include LC3B staining of *S. pneumoniae* WT vs Δ CbpC to analyse if there are differences in LC3B and/or p62 puncta after infection.

Other points:

Although the authors mention that LC3II was exclusively increased in cells expressing GFP-CbpCT4; in Fig.1D other Cbps (besides CbpCT4) also seem to increase LC3-II levels. The authors should comment and discuss on these results and why they focused on CbpC. The quality of the images in Figure 1 panel B is low, for example how many cells are in the CbpG panel?

Why the authors used a Atg3 KO in Figure 3B instead of Atg14L KO? That would have made more sense, considering the main message and claim of the paper. This should be discussed.

The panels 1G and 3K are not necessary.

There are some typos in Figure 2 ("lysete").

1st Revision - authors' response

13 January 2020

Referee #1:

Overall this is an extremely well written manuscript that proposes a novel and interesting mechanism. The experiments mapping the interaction between ATG14L and CbpC were exceptionally good. However, the conclusions and model seem premature based on the experimental data provided. This manuscript would benefit from additional experiments under conditions that are closer to physiologic to ensure the data generated from overexpression act similarly in a infection.

Response: Thank you for the positive comments.

Major Concerns:

Comment 1-1.

Fig 1. In the initial screens testing Cbp and autophagy it is not clear if overexpressed CbpC in the mammalian system is going to accurately recapitulate the effect of a cell surface proteins during infection or if bacterial autolysis releases large levels of free CbpC, enough to maintain repression of ATG14L1 levels.

*Response: We thank the reviewer for this helpful comment. As suggested by the reviewer, we examined whether bacterial autolysis releases free CbpC into the cytosol by constructing *S. pneumoniae* R6 Δ cbpF/pCbpF_{R6}-FLAG strain. As shown below (upper panels), cytosolic CbpF_{R6} signals were detected both proximal to intracellular pneumococci and free in the cytosol in cells infected with Δ cbpF/pCbpF_{R6}-FLAG. These signals were completely abolished in cells infected with *S. pneumoniae* R6 Δ cbpF. These results clearly show that bacterial autolysis releases free CbpF_{R6} into the cytosol with settings closer to physiological conditions. We have added these data to our revised manuscript. Please see Figure 1G and EV1D and E.*

We next conducted IP experiments as shown below (lower panels). We found that intracellular CbpF released from pneumococci can interact with Atg14 during infection. We added these data to our revised manuscript. Please see Figure 2F.

Comment 1-2.

Additionally, while survival was analyzed (1F) two controls were missing that I would consider critical. 1) the addition of a survival assay in an autophagy deficient background should be done. If the loss of CbpC results in higher autophagic killing then the viability should be the same in an autophagy deficient background,

Response: As suggested by the reviewer, we conducted intracellular survivability assays using Atg5-KO MEF cells and confirmed that the viability of *S. pneumoniae* R6 WT and $\Delta cbpF$ were not significantly different in an autophagy-deficient background. We add these data to our revised manuscript. Please see Figure EV3B.

Comment 1-3.

2) the model proposed would infer that cells that were not infected should not have an increase in autophagy. Given the MOI and cell type used one would expect that less than 100% of cells should have been infected.

Response: *As suggested by the reviewer, we examined the efficiency of *S. pneumoniae* R6 WT and $\Delta cbpF$ invasion into MEF cells and found that approximately 1 bacterium internalized each cell, suggesting that most cells were evenly infected. We added these data to our revised manuscript. Please see Figure EV3A.*

Comment 1-4.

Immunofluorescent staining of autophagy and bacterial markers should be performed.

Response: *As suggested by the reviewer, we examined the intracellular localization of free CbpC in *S. pneumoniae*-infected cells. As described in our response to comment 1-1, free CbpC was detected in the cytosol. We next tried to detect co-localization of cytosolic CbpC with p62 or LC3B puncta in *S. pneumoniae*-infected cells; however, we barely detected it under our experimental conditions. Because cytosolic CbpC forms puncta in *S. pneumoniae*-infected cells, and CbpC-p62 interaction and autophagic activity are essential for Atg14 degradation caused by *S. pneumoniae* infection (please see our response to comment 3-1 below), we believe that the interaction of CbpC with p62 is essential for autophagy induction and subsequent Atg14 degradation. We are*

assuming that complex formation between CbpC and LC3B or p62 might be extremely transient in *S. pneumoniae*-infected cells.

Comment 1-5.

Confusingly, based on data from later figures (p62 clearance), it seems that the majority of cells must be degrading ATG14L in response to SP, which would indicate an infection rate of ~80% or more. Therefore, it would be good to show that the autophagic response that is being measured throughout is due to the CbpC release in infected cells.

Response: We thank the reviewer for this helpful comment. As mentioned above, under our experimental conditions, the number of internalized bacteria per cell was approximately 1, suggesting that most cells were infected with at least one *S. pneumoniae* bacterium. Furthermore, as suggested by reviewer, we examined whether Atg14-degradation was due to CbpC release in infected cells. We constructed an invasion-deficient mutant ($\Delta cbpA$), an endosomal damage-deficient mutant (Δply), and a bacterial autolysis-dampened mutant ($\Delta lytA$) and examined Atg14 disappearance in *S. pneumoniae*-infected A549 cells. As shown below, Atg14 disappearance in $\Delta cbpA$, Δply , or $\Delta lytA$ -infected cells decreased dramatically to a similar level to that in $\Delta cbpF$ -infected cells, suggesting that bacterial invasion, endosomal damage, and bacterial autolysis play essential roles in Atg14 degradation in *S. pneumoniae*-infected cells. We added these data to our revised manuscript. Please see Figure EV3C and D.

Furthermore, we examined polyclonal *S. pneumoniae* invasion using a co-culture system. *S. pneumoniae* invasion-permissive A549 cells (marked with GFP) and *S. pneumoniae* invasion-non-permissive (pIgR knock-downed) A549 cells were cocultured at a 1:3 ratio, and Atg14-disappearance was studied. As shown below, robust Atg14 degradation occurred in *S. pneumoniae* invasion-permissive A549 cells (marked with GFP), but not in *S. pneumoniae* invasion-non-permissive A549 cells (pIgR knocked down). These results clearly show that bacterial invasion was essential for *S. pneumoniae*-induced Atg14 degradation. We added these data to our revised manuscript. Please see Figure 3G and H.

Comment 2-1.

Fig. 2. The interaction of ATG14 and CbpC was never shown endogenously. ATG14 was not shown to localize to invading SP.

Response: As suggested by the reviewer, we conducted IP experiments in $\Delta cbpF$ complemented with CbpF-FLAG-infected cells to detect endogenous Atg14; however, we could not detect it. We then conducted IP experiments using MEF cells stably expressing HA-Atg14 and $\Delta cbpF$ complemented with CbpF-FLAG. As shown below, the interaction of CbpF-FLAG released from pneumococci and HA-Atg14 during infection was clearly detected. We added these data to the revised manuscript. Please see Figure 2F. Because autophagy against *S. pneumoniae* is markedly suppressed by Atg14

knockdown or 3-MA (a PIK3C3 inhibitor) treatment, Atg14 is undoubtedly essential for *S. pneumoniae*-induced autophagy. However, it was difficult to observe Atg14 recruitment around intracellular pneumococci or pneumococci-containing autophagosomes. We speculate that the recruitment of PIK3C3 or Stx17–Atg14–SNAP29 complexes to pneumococci-containing vacuoles or autophagosomes is extremely transient. During xenophagy, Atg14 recruitment to bacteria has been reported in GAS Δ NADase mutant and *Salmonella*; however, this was difficult to observe in our system because the detection of each autophagy marker around intracellular pathogens depends on the types of bacteria and cells used experimentally.

Comment 2-2.

The overexpression ATG14 may well have thrown off the stoichiometry of ATG14-beclin. Which calls into question if the CbpC would be able to compete for ATG14 binding to Beclin-1. Since only small amounts of ATG14 are free of Beclin-1 (Kim et al Cell 2013) it is unclear based on the data shown if this interaction would occur under physiologic conditions.

Response: We thank the reviewer for this helpful comment. As suggested by the reviewer, we measured direct interactions between Beclin1 and Atg14 in the presence or absence of CbpC_{T4} in living cells using nano-bioluminescence resonance energy transfer (NanoBRET). We found a slight effect of CbpC on the Beclin1–Atg14 interaction (please see below, right panel). Furthermore, we confirmed the effect of CbpC on the Beclin1–Atg14 interaction by performing IP experiments. As shown below (right panel), CbpC did not compete with the Atg14–Beclin1 interaction. Rather,

CbpC_{T4}-Atg14-Beclin1 complex formation was observed. Based on these findings, we concluded that CbpC did not compete with the Atg14-Beclin1 interaction. Intriguingly, we found that Beclin1 might suppress CbpC-induced Atg14 degradation. This result also supports our notion that CbpC can manipulate Atg14-Stx17 interactions via Atg14 degradation without adversely affecting Beclin1-Atg14-based PIK3C3 activity. We added these data to the revised manuscript. Please see Figure EV3G and H.

Comment 3-1.

Fig. 3. The loss of ATG14 (esp as viewed by IF is profound). The only link between ATG14 loss and autophagy is in Fig. 3B. However, in the delta-CbpC mutant ATG14 levels at lower in WT control, then higher at 3 hours. Effectively we have one lane that shows CbpC dependence and one that does not. No controls showing other cellular responses were not affected and that similar bacterial load was added. The rescue of ATG14 clearance in an autophagy deficient line or through pharmacological inhibitors, but not through the additional of proteasomal inhibitors would show the dependence on autophagy for ATG14 degradation.

Response: *We thank the reviewer for this helpful comment. As suggested by the reviewer, we examined the inhibitory effect of bafilomycin on CbpC-induced Atg14 degradation in cells infected with S. pneumoniae WT or ΔcbpC. As shown below (upper panel), Atg14 degradation in WT-infected cells was dramatically recovered by*

bafilomycin treatment. We substituted our original figure with this one in our revised manuscript. Furthermore, we examined whether atg5 or p62 knockdown could suppress Atg14 degradation in IF experiments, using A549 cells. As shown below (lower panel), p62 knockdown dramatically suppressed the disappearance of perinuclear Atg14 caused by S. pneumoniae infection. We also found that the suppressive effect of atg5 knockdown on the disappearance of perinuclear Atg14 was not as strong as that observed in p62 knock-down cells, implying the partial involvement of an Atg5-independent degradative pathway, such as the ubiquitin–proteasome system. We added these results to the revised manuscript. Please see Figure 3B and K.

Furthermore, we constructed S. pneumoniae R6 $\Delta cbpF$ derivatives complemented with a series of p62 interaction-deficient CbpF_{R6} variants via homologous recombination, and Atg14-degradation assays were performed in A549 cells using these strains. As shown below, the disappearance of perinuclear Atg14 caused by S. pneumoniae

infection decreased dramatically in cells infected with p62 binding-deficient, CbpF_{R6}-expressing *S. pneumoniae* strains. This result also supports our notion that the Atg14–CbpC–p62 interaction plays a pivotal role in Atg14 degradation. We added these data to the revised manuscript. Please see Figure 5H and I, and EV5G.

Comment 3-2.

The IF data is strong; however, as with other figures there is no evidence that ATG14 is stable in cells that were not infected as would be predicted by the model. Perhaps a lower MOI could be used leaving a polyclonal population linking bacterial internalization to ATG14 loss.

Response: As suggested by the reviewer, we constructed a co-culture system to make polyclonal bacterial internalization possible. As described in the response to comment 1-5, Atg14 degradation was clearly dependent on *S. pneumoniae* invasion into the cells (please see figure below, upper panel). Furthermore, as also mentioned in our response

to comment 1-5, bacterial invasion, endosomal damage, and bacterial autolysis play pivotal roles in Atg14 degradation (please see below, lower panel).

Minor concerns:

- Model Fig. 1G would be more informative if a distinction was made between autophagy and xenophagy (or it could be removed)

Response: As suggested by the reviewer, the model originally shown in Fig. 1G was removed.

• ATG14L was the old nomenclature, ATG14 is the current name (capitalization for proteins, such as ATG14 should also follow standard conventions https://www.jci.org/kiosk/publish/genestyle)

***Response:*** *We have corrected the nomenclature, as suggested by the reviewer.*

• Great mapping in Figure 4. Would have been nice to solidify some of the key previous data (ATG14 loss) with this mutant, underscoring the importance of the interaction

***Response:*** *We thank the reviewer for this insightful comment. As described in our response to comment 3-1, we constructed S. pneumoniae R6 ΔcbpF derivatives complemented with a series of p62 interaction-deficient* $CbpF_{R6}$ *variants via homologous recombination, and then Atg14-degradation assays were performed in A549 cells using these strains. We found that the disappearance of perinuclear Atg14 caused by S. pneumoniae infection decreased dramatically in cells infected with p62 binding-deficient,* $CbpF_{R6}$*-expressing S. pneumoniae strains. This result clearly shows that the Atg14–CbpC–p62 complex was indispensable for Atg14 degradation.*

R6 $\Delta cbpF$:CbpF_{R6}-HA

- Ub ligase activity of TRAF6 could have been tested or discussed in figure 5 as a modulator of complex formation

Response: We thank the reviewer for this helpful comment. As suggested by the reviewer, we examined the dependence of TRAF6 E3 ligase activity in promoting the p62–CbpC interaction and found that TRAF6 E3 ligase activity was dispensable for facilitating the p62–CbpC interaction. We replaced Figure EV5D with a new version showing the result with Traf6 C70A.

Referee #2:

This is a very interesting manuscript. Clearly autophagy is a relatively new area of study for pathogens that invade so this work is timely. Recent work from this group indicated that when pneumococci enter the cytoplasm, pneumolysin participates in directing the bacteria towards the autophagy pathway for degradation. This new manuscript describes a second safe path through cells. The existence of this second path fits with decades of

work indicating that pneumococci can traverse endothelial and epithelial barriers by passing safely through cells (no indication of barrier breakdown). Thus, it is an important mechanistic study that establishes some details about an important step in invasive disease. The role of CbpC is a surprise that adds further spice and intrigue to the mechanism. This paper demonstrates the biochemistry of the CbpC interaction with autophagy components in model systems

Response: Thank you for the positive comments.

The experiments are clearly presented with excellent controls. Thus, the hypotheses are supported by excellent data in model systems. The only comment is that all of the systems used are not related to cells pneumococci encounter during disease. It remains to be seen how this mechanism plays out in cells that represent the actual sites of transmigration during infection. Could the authors add images of CbpC and autophagy components colocalizing on bacteria crossing endothelial or epithelial barriers?

Response: Thank you for the positive comment. In the original and revised versions of the manuscript, we used human pulmonary epithelial cell-derived A549 cells to investigate Atg14 degradation during S. pneumoniae infection. We plan to examine the effect of CbpC on autophagic activity in endothelial cells in a future project. As suggested by the reviewer, we examined the intracellular localization of free CbpC in S. pneumoniae-infected cells. As shown below, cytosolic CbpF_{R6} signals were detected proximal to intracellular pneumococci and free in the cytosol in cells infected with S. pneumoniae R6 Δ cbpF/pCbpF_{R6}-FLAG. These signals were completely abolished in Δ cbpF-infected cells. We added these data to the revised manuscript. Please see Figure 1G and EVID and E.

Because autophagy against *S. pneumoniae* is markedly suppressed by Atg14 knockdown or 3-MA (a PIK3C3 inhibitor) treatment, Atg14 is undoubtedly essential for *S. pneumoniae*-induced autophagy. However, we failed to observe the recruitment of Atg14 around intracellular pneumococci or pneumococci-containing autophagosomes. We speculate that the recruitment of PIK3C3 or Stx17–Atg14–SNAP29 complexes to pneumococci-containing vacuoles or autophagosomes is extremely transient. During xenophagy, Atg14 recruitment to bacteria was reported in a GAS Δ NADase mutant and in *Salmonella*. However, it was difficult to observe under our experimental conditions because the detectability of each autophagy marker around intracellular pathogens depends on the types of bacteria and cells used under the experimental settings.

Therefore, we further investigated the importance of the CbpC–p62 interaction and autophagic activity in terms of Atg14 degradation in A549 cells infected with *S. pneumoniae* (please see below). Taken together, we believe that the CbpC–p62 interaction is undoubtedly essential for autophagy induction and subsequent Atg14 degradation in human pulmonary epithelial cells. We added these results to the revised manuscript. Please see Figure 3K, 5H and I, and EV5G.

Referee #3:

Overall, the study is well performed and well written. There is no much known about the interactions between *S. pneumoniae* and autophagy and this study provides a major contribution to this topic. However, there are some major points that are unclear and some experiments needed to confirm the findings using the bacteria.

Response: Thank you for the positive comment.

Major points:

1. The data with the expression of CbpC, including the dp3 and dp5 CbpCT4 mutants should be complemented with experiments showing this is valid for the infection with *S. pneumoniae*. Not sure if all of these mutants are available but at least some infections with *S. pneumoniae* and show targeting either to autophagosomes or lysosomes. This is important to confirm the interactions reported here are also valid during infection of cells by *S. pneumoniae*.

*Response: We thank the reviewer for this insightful comment. As suggested, we constructed *S. pneumoniae* R6 Δ cbpF derivatives complemented with a series of p62 interaction-deficient cbpF_{R6} variants via homologous recombination, and then Atg14-degradation assays were performed in A549 cells using these strains. As shown below, we found that the disappearance of perinuclear Atg14 caused by *S. pneumoniae* infection decreased dramatically in cells infected with p62 binding-deficient, CbpF_{R6}-expressing *S. pneumoniae* strains. We added these data to the revised manuscript. Please see Figure 5H and I, and EV5G.*

R6 $\Delta cbpF$:CbpFR6-HA

2. In this context, Fig.3 should also include LC3B staining of *S. pneumoniae* WT vs $\Delta CbpC$ to analyse if there are differences in LC3B and/or p62 puncta after infection.

Response: We thank the reviewer for this helpful comment. As suggested by reviewer, we examined the intracellular localization of free CbpC by constructing *S. pneumoniae* R6 $\Delta cbpF/pCbpFR6-FLAG$ strain. As shown below in the upper panels, cytosolic CbpFR6 signals were detected both proximal to intracellular pneumococci and free in the cytosol in cells infected with $\Delta cbpF/pCbpFR6-FLAG$. These signals were completely abolished in cells infected with *S. pneumoniae* R6 $\Delta cbpF$. We added these data to the revised manuscript. Please see Figure 1G and EV1D and E.

We next studied the co-localization of cytosolic CbpC with LC3B or p62 puncta in *S. pneumoniae*-infected cells, however, such co-localization was barely detectable under our experimental conditions. Because cytosolic CbpC forms puncta in *S.*

pneumoniae-infected cells, and CbpC–p62 interaction and autophagic activity are essential for Atg14 degradation caused by *S. pneumoniae* infection (please see our response to comment 1 and the lower panel in the figure below), we believe that the interaction of CbpC with p62 is essential for autophagy induction and subsequent Atg14 degradation. We are assuming that complex formation between CbpC and LC3B or p62 might be extremely transient in *S. pneumoniae*-infected cells.

Other points:

1. Although the authors mention that LC3II was exclusively increased in cells expressing GFP-CbpCT4; in Fig.1D other Cbps (besides CbpCT4) also seem to increase LC3-II levels. The authors should comment and discuss on these results and why they

focused on CbpC. The quality of the images in Figure 1 panel B is low, for example how many cells are in the CbpG panel?

Response: We apologize for the confusion and we thank the reviewer for this helpful comment. We quantified the LC3-II levels in Fig. 1D. As shown below, the LC3-II levels exclusively increased in cells expressing GFP-CbpC (please see below, left panel). We also apologize for the confusion regarding Fig. 1B. The CbpG panel includes two cells. We improved the image qualities, indicated each individual cell with a dotted line, and changed the CbpF_{T4} and CbpL panels (please see below, right panel). We added these data to the revised manuscript. Please see Figure 1B, D and EV1A.

2. Why the authors used a Atg3 KO in Figure 3B instead of Atg14L KO? That would have made more sense, considering the main message and claim of the paper. This should be discussed.

Response: We apologize for the confusion. In Fig. 3B, we examined autophagy-dependent Atg14 degradation in cells infected with *S. pneumoniae* WT or Δ cbpC. Therefore, we could not use Atg14-KO MEF cells. As an alternative, we examined the inhibitory effect of bafilomycin on Atg14 degradation in *S. pneumoniae* WT or Δ cbpC infected cells. As shown below, Atg14 degradation in *S. pneumoniae*

WT-infected cells was dramatically recovered by bafilomycin treatment. We substituted the original data with these data. Please see Fig. 3B in the revised manuscript.

3. The panels 1G and 3K are not necessary.

Response: As suggested by the reviewer, we removed panel 1G. We believe that panel 3K can help readers understand the manuscript. Thus, we simplified the figure and moved it to EV3I in the revised manuscript.

4. There are some typos in Figure 2 ("lysete").

Response: Thank you for this comment. We have corrected this misspelling.

2nd Editorial Decision

10 February 2020

Thank you for the submission of your revised manuscript to our editorial offices. We have now received the reports from the three referees that were asked to re-evaluate your study, you will find below. As you will see, the referees now fully support the publication of your study in EMBO reports.

Before we can proceed with formal acceptance, I have these editorial requests I ask you to address in a final revised version of the manuscript:

- Please provide the abstract written in present tense throughout.
- Please provide individual production quality figure files as .eps, .tif, .jpg (one file per figure), of main figures and EV figures. Please upload these as separate, individual files upon re-submission.
- In the figure legends (main and EV figures) you state several time that the data shown were obtained 'from three independent experiments'. Please indicate in each case if these were biological or technical replicates.
- Please carefully check that each single figure panel is called out in the text. Presently, it seems that several panels of Fig. 3 and panels 5D and 5E are not called out.
- Please enter the complete funding information also into our submission system (including grant numbers). Please check that in the online form and the manuscript the funding information is the same and complete.
- As some of the Western blots shown are significantly cropped, could you provide the source data for all the Western Blot images? The source data will be published in a separate source data file online along with the accepted manuscript and will be linked to the relevant figure. Please submit the source data (scans of entire gels or blots) together with the final revised manuscript. Please include size markers for the scans of entire gels, label the scans with figure and panel number, and send one PDF file per figure (main and EV figures).

In addition I would need from you:

- a short, two-sentence summary of the manuscript
- two to three bullet points highlighting the key findings of your study
- a schematic summary figure (in jpeg or tiff format with the exact width of 550 pixels and a height of not more than 400 pixels) that can be used as a visual synopsis on our website.

Referee #1:

The authors have done a tremendous amount of work including key experiments and have adequately addressed my original concerns.

Referee #2:

All questions have been answered thoroughly.

Referee #3:

The authors have satisfactorily addressed all my concerns. I recommend publication in EMBO reports.

2nd Revision - authors' response

28 February 2020

The authors performed all minor editorial changes.

Corresponding Author Name: Michinaga Ogawa

Manuscript Number: EMBOR-2019-49232V1